# MULTI-PREFERENCE OPTIMIZATION: GENERALIZING DPO VIA SET-LEVEL CONTRASTS

## ABSTRACT

Practical post-training pipelines involving on-policy generation typically produce multiple candidate responses per prompt. However, popular alignment methods like Direct Preference Optimization (DPO) are restricted to pairwise comparisons, discarding valuable supervisory signal. In this setting, we propose Multi-Preference Optimization (MPO), a generalization of DPO that optimizes over entire sets of selected and rejected responses. This set-level contrastive approach is theoretically grounded: we first prove that leveraging $n$ responses achieves a $\mathcal{O}\left(\frac{1}{\sqrt{n}}\right)$ convergence in TV-distance to the true preference distribution. We then prove, under a formal model with spacing-scaled Gaussian noise ($\Delta, \sigma = \mathcal{O}(1/n)$), that MPO's 2-bin partition reliability remains bounded away from zero, in contrast to full-ranking methods which degrade exponentially ($\exp(-\mathcal{O}(n))$). To further enhance learning, MPO employs a deviation-based weighting, which emphasizes outlier responses to induce an implicit curriculum. Empirically, as we show over multiple models and benchmarks, MPO achieves state-of-the-art performance, with an improvement of up to $\sim 17.5\%$ WR on AlpacaEval2 in the on-policy iterative setting, and state-of-the-art results in off-policy settings.

## 1 INTRODUCTION

Aligning large language models (LLMs) to follow instructions effectively and adhere to desired behaviors is a cornerstone of modern AI development (Ouyang et al., 2022; Christiano et al., 2017). Direct Preference Optimization (DPO) has emerged as a prominent and computationally efficient paradigm for the alignment task. DPO bypasses the need for explicit reward model training by directly optimizing a policy based on pairwise preferences (Rafailov et al., 2024). However, the standard DPO framework is inherently limited to comparing single pairs of (preferred, dispreferred) responses; furthermore, the standard DPO implementations use the most-preferred and the least-preferred responses per query (Wu et al., 2024; Meng et al., 2024) discarding the rest.

This pairwise formulation becomes restrictive as modern alignment pipelines, especially those involving on-policy generation or self-play, often produce a multitude of candidate responses per prompt (Wu et al., 2024; Chen et al., 2024c; Pang et al., 2024; Tang et al., 2025), wasting valuable supervisory signal and computational effort. To overcome this, we propose **Multi-Preference Optimization (MPO)**, which generalizes DPO by using a group-contrastive loss to compare entire sets of "accepted" versus "rejected" responses. This core idea extends DPO's underlying Bradley-Terry model (Bradley & Terry, 1952) from pairwise to groupwise comparisons.

Leveraging multiple responses for alignment is by no means new. Several alternative strategies have been proposed for post-training optimization using richer preference data. Traditional policy-gradient RLHF methods, such as Proximal Policy Optimization (PPO) (Schulman et al., 2017), can implicitly utilize multiple responses through a learned reward function. More recent methods explicitly target groupwise comparisons, like Group Relative Policy Optimization (GRPO) (Liu et al., 2024a), or aim to integrate explicit reward signals more directly, including Reward-Aware Preference Optimization (RPO) (Sun et al., 2025) and cross-

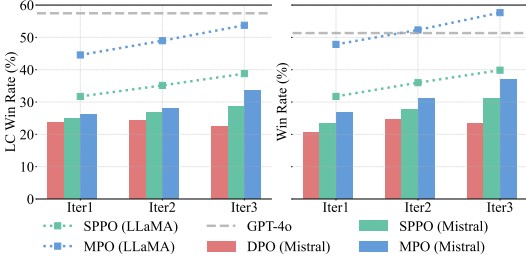

Figure 1: Multi-Preference Optimization (MPO) achieves state-of-the-art results on WR, LC-WR on AlpacaEval2, with performance nearly matching GPT-4o, demonstrating MPO's efficacy for post-training optimization.

Figure 2: Mᴘᴏ pipeline: Responses are split into accepted and rejected sets using human labels or reward model scores. We assign weights based on deviation from the mean and optimize to upweight the accepted group while downweighting the rejected group using our Mᴘᴏ loss.

entropy-based methods like InfoNCA (Chen et al., 2024a). In fact, we could apply the standard DPO loss across all possible $\binom{k}{2}$ pairs (Tunstall et al., 2023), without discarding any of the $k$ preferences per query; or use ranking-based methods to leverage full or partial rankings over all $k$ responses (Dong et al., 2023; Yuan et al., 2023).

Many of these approaches, however, face significant hurdles. Methods like PPO, and RPO typically necessitate a well-calibrated reward model to provide the scalar values or advantages crucial for their optimization objectives (Schulman et al., 2017; Liu et al., 2024a; Sun et al., 2025). In fact methods like GRPO only work with verifiable rewards (Liu et al., 2024a). Similarly, InfoNCA relies on the magnitude of explicit reward scores for its target distribution (Chen et al., 2024a). The challenge of obtaining consistently accurate reward scores, particularly given the known difficulties in training stable value functions via regression (Farebrother et al., 2024), can make these approaches sensitive to reward noise or mis-calibration. On the other hand, naively applying DPO to all pairs incurs significant computational overhead and potential optimization instabilities, while Plackett-Luce models demand reliable full rankings, which are often expensive and difficult to obtain (Chen et al., 2024b), and may cause alignment brittleness.

Mᴘᴏ is designed to overcome these specific challenges by retaining DPO's robust contrastive nature while effectively scaling to multiple preferences. It does not need well-calibrated reward models or full ranking over the responses, and instead can work with relative comparisons (ordinal preferences) over exact value regression (Farebrother et al., 2024). When fine-grained scores are available, these sets are partitioned into "accepted" and "rejected" groups based on a query-specific score mean. Mᴘᴏ then employs a novel groupwise contrastive loss, generalizing the Bradley-Terry model (Bradley & Terry, 1952), which is distinctively enhanced by a deviation-based weighting scheme. This weighting prioritizes informative outliers, creating an implicit curriculum (Bengio et al., 2009) and using reward scores primarily to define groups and modulate influence. Mᴘᴏ provides a theoretically grounded method with state-of-the-art results (Section 7).

We provide a detailed illustration of our method in Figure 2. Our primary contributions are:

**1. Generalising Multi-Preference Optimization (Mᴘᴏ):** We introduce Multi-Preference Optimization (Mᴘᴏ), which generalizes DPO's pairwise contrast to efficiently learn from *sets* of responses. Mᴘᴏ employs a novel, theoretically-grounded groupwise objective. We prove (Thm. 2) this set-wise approach is provably robust to reward model noise, while enabling richer supervision than single positive/negative pairs (Thm. 1).

**2. Advantage based Variant (W-Mᴘᴏ):** We propose Weighted MPO (W-Mᴘᴏ), an extension that incorporates fine-grained reward information by additively adjusting response logits based on their deviation from the query-specific mean reward. This prioritizes informative outliers, creating an implicit data-driven curriculum (Section 4.4).

**3. State-of-the-Art Alignment Performance:** Both Mᴘᴏ and W-Mᴘᴏ achieve state-of-the-art results on diverse alignment benchmarks, including AlpacaEval 2.0, Arena-Hard, and MT-Bench, outperforming established baselines with various LLMs (e.g., Mistral-7B and Llama-3-8B).

## 2 Related Work

We briefly situate Mᴘᴏ within the evolving landscape of preference optimization, deferring a more comprehensive survey to Appendix B. The alignment of Large Language Models (LLMs) has significantly advanced from early Reinforcement Learning from Human Feedback (RLHF) frameworks, which typically employ reward models and policy gradient methods like Pᴘᴏ (Schulman et al., 2017; Ouyang et al., 2022). A pivotal shift occurred with Direct Preference Optimization (Dᴘᴏ) (Rafailov et al., 2024), which directly learns from pairwise preferences using a contrastive loss based on the

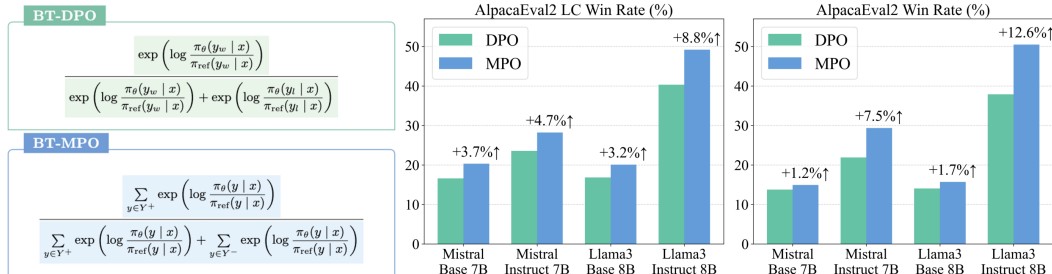

Figure 3: **MPO generalizes DPO** by extending the Bradley-Terry formulation to groupwise comparisons over multiple responses. This richer supervision yields stronger performance across model families. MPO gracefully reduces to DPO when each set contains a single response.

Bradley-Terry model (Bradley & Terry, 1952), simplifying the alignment pipeline. This contrastive learning paradigm, where models learn by distinguishing between positive and negative examples, has foundations in self-supervised learning, notably with objectives like InfoNCE (Oord et al., 2018) which maximize mutual information between related samples.

Many recent works extend DPO's pairwise approach. Some focus on diversified objectives or reward modeling, such as KTO (Ethayarajh et al., 2024) and TDPO (Zeng et al., 2024) for response or token-level alignment. Others modify the DPO framework by altering the reference model, adding regularizers for issues like length bias, or unifying preference learning with supervised objectives, exemplified by SPIN (Chen et al., 2024c), CPO (Xu et al., 2024), ORPO (Hong et al., 2024), SimPO (Meng et al., 2024), R-DPO (Park et al., 2024), and LD-DPO (Liu et al., 2024c). Other works explore different structures for preference signals beyond simple pairs, such as list-wise or rank-based supervision as seen in RAFT (Dong et al., 2023) and RRHF (Yuan et al., 2023).

Leveraging the multiple responses common in on-policy generation or rich datasets (Cui et al., 2023; Gupta et al., 2025) necessitates moving beyond pairwise DPO. Distribution-matching methods like InfoNCA (Chen et al., 2024a) deal with multiple responses by replicating a target reward distribution via a KL-divergence style objective. On the other hand, MPO provides a distinct, set-level *contrastive* objective that generalizes the Bradley-Terry choice model, aiming only to distinguish preferred from dispreferred sets (see Appendix E for a full gradient analysis). This set-based comparison robustly utilizes group-level preference signals for post-training optimization.

## 3 PRELIMINARIES AND TECHNICAL BACKGROUND

Our goal is to learn a policy $\pi_\theta$ based on feedback on the candidate responses $\{y_1, \ldots, y_n\}$ to a query $x$. This feedback often implies a "strength" or "utility" $u(y)$ for each response. This utility $u(y)$ is derived from the policy's log-probabilities relative to a reference model $\pi_{\text{ref}}$, specifically $u(y) = \exp(\beta r_\theta(y \mid x))$, where $r_\theta(y \mid x) = \log(\pi_\theta(y \mid x)/\pi_{\text{ref}}(y \mid x))$ and $\beta$ is a positive scaling factor. Different alignment methods incorporate these utilities based on how responses are grouped (or "binned") and how preferences between these responses are modeled. Broadly, these approaches rest on a shared conceptual foundation:

> **A Unified View of Preference Optimization**
>
> *Preference optimization can be conceptualized as a probabilistic selection process. The likelihood of favoring certain responses over others is governed by underlying "strength" or "utility" measures associated with each response by a policy model $\pi_\theta$ being aligned. We utilize this core framework that underlies both Choice theory and Bradley-Terry models to design a reward function for LLM policy optimization.*

### 3.1 PAIRWISE AND SETWISE PREFERENCE MODELS (DPO AND MPO)

The foundation for modeling preferences between groups of items can be traced to choice theory (Luce et al., 1959). We consider two disjoint sets of responses: a preferred ("chosen") set $\mathcal{Y}^+$ and a dispreferred ("rejected") set $\mathcal{Y}^-$. Following the Bradley-Terry model, we assume each response $y$ has an intrinsic utility $u(y)$. The probability of a single response drawn from the combined pool $\mathcal{Y}_{\text{all}} = \mathcal{Y}^+ \cup \mathcal{Y}^-$ belonging to the preferred set $\mathcal{Y}^+$ is then given by:

$$P(\mathcal{Y}^+ > \mathcal{Y}^-) = \frac{\sum_{y \in \mathcal{Y}^+} u(y)}{\sum_{y' \in \mathcal{Y}_{\text{all}}} u(y')} = \frac{\sum_{y \in \mathcal{Y}^+} u(y)}{\sum_{y \in \mathcal{Y}^+} u(y) + \sum_{y \in \mathcal{Y}^-} u(y)}. \tag{1}$$

In the context of preference optimization and following the Bradley-Terry model (Bradley & Terry, 1952), these utilities $u(y)$ are often defined as an exponential function of an underlying score or logit $s_y$, i.e., $u(y) = \exp(s_y)$. This exponential transformation ensures positivity of utilities and naturally leads to the logistic (sigmoid) function when comparing two individual items. Eq. 1 forms the conceptual basis for both DPO and our proposed MPO (See comparison in Figure 3).

**Direct Preference Optimization (DPO).** DPO(Rafailov et al., 2024) is a special case of Eq. 1 where each set contains a single item: $\mathcal{Y}^+ = \{y_w\}$ (winner) and $\mathcal{Y}^- = \{y_l\}$ (loser). The DPOobjective is to maximize $\log P(y_w > y_l)$. Using $u(y) = \exp(\beta r_\theta(y|x))$ implies $\log u(y) = \beta r_\theta(y|x)$, so the standard DPO loss becomes:

$$\mathcal{L}_{\text{DPO}} = -\mathbb{E}_{(x,y_w,y_l)} \big[ \log \sigma \big( \beta(r_\theta(y_w \mid x) - r_\theta(y_l \mid x)) \big) \big]. \tag{2}$$

**Multi-Preference Optimization (MPO): A generalization of DPO.** Our work, MPO, directly utilizes the setwise preference model from Eq. 1 where $\mathcal{Y}^+$ and $\mathcal{Y}^-$ can contain multiple responses. The MPO objective is to maximize $\log P(\mathcal{Y}^+ > \mathcal{Y}^-)$, using the same utility definition $u(y) = \exp(\beta r_\theta(y|x))$. The specific construction of $\mathcal{Y}^+, \mathcal{Y}^-$ from data, and the introduction of weighted utilities for the W-MPO variant, are detailed in Section 4.

### 3.2 RANK-BASED AND DISTRIBUTIONAL PREFERENCE MODELS

**Plackett-Luce (PL) Model.** When a full ranking (permutation) $y_{(1)} \succ y_{(2)} \succ \cdots \succ y_{(n)}$ over $n$ items is available, the PL model (Plackett, 1975) defines its probability as $P_{\text{PL}}(\text{ranking}) = \prod_{i=1}^n \frac{u(y_{(i)})}{\sum_{j=i}^n u(y_{(j)})}$, where $u(\cdot)$ represents item utilities. This can be conceptualized as sequentially selecting items for $n$ ordered "bins" (ranks), where the probability of selecting an item for the current rank depends on the utilities of the remaining unranked items. PL thus models fine-grained ordinal relationships, contrasting with the binary set partition used in MPO.

**Distribution Matching (e.g., InfoNCA).** InfoNCA (Chen et al., 2024a) and similar methods align $\pi_\theta$ by matching its output distribution (derived from utilities $u(y_i)$) to a target distribution. For $n$ responses $\{y_i\}$ with scalar quality scores $S_i$, a target probability $p_{\text{target}}(y_i) = \text{softmax}_i(S_i/\tau)$ is often formed. The model's predicted probability for $y_i$ is $p_{\text{model}}(y_i) = \text{softmax}_i(\log u(y_i))$. The objective then is to minimize the cross-entropy:

$$\mathcal{L}_{\text{InfoNCA}} \propto -\sum_{i=1}^n p_{\text{target}}(y_i) \log p_{\text{model}}(y_i). \tag{3}$$

This loss function aims to replicate a reward distribution over individual responses.

## 4 MPO: MULTI-PREFERENCE OPTIMIZATION

Building upon the principles of groupwise preference modeling discussed in Section 3, we introduce Multi-Preference Optimization (MPO), a method designed to align language models using multiple preferred and dispreferred responses per query. MPO generalizes DPO's pairwise comparison to a groupwise contrast using multiple preferred/dispreferred responses per query, enabling richer supervision. We now define the MPO objective and its weighted W-MPO variant.

### 4.1 PROBLEM SETUP FOR MPO

For the MPO framework, we operate on a dataset $\mathcal{D}_{\text{MPO}}$. Each instance in this dataset consists of a query $x$, a set of $n_x^+$ chosen (preferred) responses $\mathcal{Y}_x^+ = \{y_{c,1}, \ldots, y_{c,n_x^+}\}$, and a disjoint set of $n_x^-$ rejected (dispreferred) responses $\mathcal{Y}_x^- = \{y_{r,1}, \ldots, y_{r,n_x^-}\}$. These sets $\mathcal{Y}_x^+$ and $\mathcal{Y}_x^-$ can be sourced from direct annotations or derived from finer-grained scalar reward scores, as elaborated for the W-MPO variant (Section 4.4). Our objective is to fine-tune the policy $\pi_\theta(y \mid x)$ using the reference model $\pi_{\text{ref}}(y \mid x)$ to define the implicit log-preference scores $r_\theta(y \mid x)$ (as introduced in Preliminaries, Eq. 3), which are scaled by an inverse temperature $\beta > 0$.

## 4.2 Base Mpo Objective

Mpo leverages sets of preferences by modeling the probability $P(\mathcal{Y}_x^+ > \mathcal{Y}_x^- \mid x; \theta, \beta)$ using the groupwise choice principle (Eq. 1 from Preliminaries). For Mpo, the utility $u(y)$ of a response $y$ is defined as $u(y) = \exp(\beta \cdot r_\theta(y \mid x))$, where $r_\theta(y \mid x)$ is the implicit log-preference score (Eq. 3). Substituting this utility definition into the general groupwise model (Eq. 1) directly yields the MPO preference probability:

$$P(\mathcal{Y}_x^+ > \mathcal{Y}_x^- \mid x; \theta, \beta) = \frac{\sum_{y \in \mathcal{Y}_x^+} \exp(\beta \cdot r_\theta(y \mid x))}{\sum_{y' \in \mathcal{Y}_x^+ \cup \mathcal{Y}_x^-} \exp(\beta \cdot r_\theta(y' \mid x))}. \tag{4}$$

The Mpo objective is to maximize this probability by minimizing its negative log-likelihood:

$$\mathcal{L}_{\text{MPO}}(\theta) = -\mathbb{E}_{(x, \mathcal{Y}_x^+, \mathcal{Y}_x^-) \sim \mathcal{D}_{\text{MPO}}} \left[ \log P(\mathcal{Y}_x^+ > \mathcal{Y}_x^- \mid x; \theta, \beta) \right]. \tag{5}$$

Intuitively, this loss function encourages the policy $\pi_\theta$ to assign collectively higher implicit preference scores $r_\theta(y \mid x)$ (and thus higher probabilities $\pi_\theta(y \mid x)$) relative to the reference policy $\pi_{\text{ref}}$) to the responses in the preferred set $\mathcal{Y}_x^+$ compared to the combined set of preferred and dispreferred responses $\mathcal{Y}_x^+ \cup \mathcal{Y}_x^-$.

## 4.3 Algorithm for Mpo

The training procedure for the core Mpo objective is summarized in Algorithm 1. The algorithm iterates through the preference dataset $\mathcal{D}_{\text{MPO}}$. For each data sample, comprising a query $x$ with its associated preferred ($\mathcal{Y}_x^+$) and dispreferred ($\mathcal{Y}_x^-$) response sets, it computes the implicit scores, then the Mpo loss term, and updates the policy model $\pi_\theta$ using gradient descent.

## 4.4 Weighted Mpo (W-Mpo) Variant

The MPO objective treats all responses within preferred ($\mathcal{Y}_x^+$) or dispreferred ($\mathcal{Y}_x^-$) sets uniformly. However, real-world preference data, often available as scalar quality scores $S(y)$, exhibit significant intra-set quality variations. This fine-grained information enables the calculation of an *advantage* for each response, $S(y) - S_{\text{mean}}(x)$. While traditional advantage-based methods like GRPO use the *signed* advantage as a direct optimization target, Weighted Mpo (W-Mpo) is de-

---

**Algorithm 1 Multi-Preference Optimization (MPO)**

1: **Input:** Initial policy $\theta_0$, $\mathcal{D}_{\text{MPO}} = \{(x, \mathcal{Y}_x^+, \mathcal{Y}_x^-)\}$, Reference policy $\pi_{\text{ref}}$, temperature $\beta$, learning-rate $\eta$
2: Initialise $\theta \leftarrow \theta_0$
3: **for** each training iteration **do**
4:     **for all** $(x, \mathcal{Y}_x^+, \mathcal{Y}_x^-) \in \mathcal{D}_{\text{MPO}}$ **do**
5:         **for** $y \in \mathcal{Y}_x^+ \cup \mathcal{Y}_x^-$ **do**
6:             $r_\theta(y \mid x) \leftarrow \log \frac{\pi_\theta(y \mid x)}{\pi_{\text{ref}}(y \mid x)}$
7:         **end for**
8:         $S_x^+ \leftarrow \sum_{y \in \mathcal{Y}_x^+} \exp(\beta \, r_\theta(y \mid x))$
9:         ▷ Selected responses
10:        $S_x^{\text{all}} \leftarrow \sum_{y \in \mathcal{Y}_x^+ \cup \mathcal{Y}_x^-} \exp(\beta \, r_\theta(y \mid x))$
11:        $L_x \leftarrow -\log(S_x^+ / S_x^{\text{all}})$
12:        Compute $\nabla_\theta L_x$
13:        $\theta \leftarrow \theta - \eta \nabla_\theta L_x$
14:     **end for**
15: **end for**
16: **return** $\theta$

---

signed to adapt this concept for the *contrastive* framework. W-Mpo leverages the *absolute advantage* (or deviation) to modulate each response's influence within the loss function. This deviation-weighting creates an implicit, data-driven curriculum by prioritizing informative outliers, which consequently dominate the gradient of the loss.

For W-Mpo, we assume query $x$ has $n_x$ responses $\mathcal{Y}_x = \{y_1, \ldots, y_{n_x}\}$ with corresponding scalar quality scores $\{S(y_i)\}_{i=1}^{n_x}$. First, we compute the mean score for the query $S_{\text{mean}}(x) = \sum_{i=1}^{n_x} S(y_i)/n_x$. Responses are then partitioned into a preferred set $\mathcal{Y}_x^+ = \{y \in \mathcal{Y}_x \mid S(y) > S_{\text{mean}}(x)\}$ and a dispreferred set $\mathcal{Y}_x^- = \{y \in \mathcal{Y}_x \mid S(y) \leq S_{\text{mean}}(x)\}$. (If $\mathcal{Y}_x^+$ is empty, the sample may be skipped). The *absolute deviation* of each response's score $S(y)$ from this mean is:

$$\Delta W_{\text{abs}}(y) = |S(y) - S_{\text{mean}}(x)|. \tag{6}$$

W-Mpo then modifies the logit for each response $y \in \mathcal{Y}_x$ by additively incorporating this absolute deviation:

$$r_\theta'(y \mid x; \alpha_w) = \beta \cdot r_\theta(y \mid x) + \alpha_w \cdot \Delta W_{\text{abs}}(y), \tag{7}$$

where $r_\theta(y \mid x)$ is the base implicit score (Eq. 3) and $\alpha_w > 0$ is a hyperparameter scaling the deviation's impact.

The additive term $\alpha_w \cdot \Delta W_{\text{abs}}(y)$ boosts the logit of outlier responses, thereby increasing their influence. The W-MPO loss function then uses these modified logits within the MPO framework:

$$\mathcal{L}_{\text{W-MPO}}(\theta) = -\mathbb{E}_{(x, \mathcal{Y}_x, \{S(y)\})} \left[ \log \left( \frac{\sum_{y \in \mathcal{Y}_x^+} \exp(r'_\theta(y \mid x; \alpha_w))}{\sum_{y' \in \mathcal{Y}_x^+ \cup \mathcal{Y}_x^-} \exp(r'_\theta(y' \mid x; \alpha_w))} \right) \right]. \tag{8}$$

The expectation is over the original responses and their scores, from which $\mathcal{Y}_x^+$ and $\mathcal{Y}_x^-$ are derived.

## 5  THEORETICAL ANALYSIS

First, we establish the formal motivation for moving beyond pairwise DPO by proving that leveraging multiple preferences ($n$) yields a provably faster convergence to the true, underlying preference distribution. We then provide justification for MPO's specific set-wise design by proving that its robustness under reward model noise is superior to that of full-ranking methods.

**1. Convergence Rate with Multiple Preferences.**   The goal of preference alignment is to learn a policy $P_{\text{learned}}$ that approximates an ideal, true preference distribution $P_{\text{true}}$. Standard DPO uses a single pairwise preference ($n = 1$). The MPO framework generalizes the DPO loss to incorporate $n > 1$ preferred responses per query. We formally prove that leveraging a larger set of $n$ preferences yields a provably tighter approximation of the ideal alignment policy, as quantified by the convergence rate of the Total Variation (TV) distance.

**Theorem 1** (**Convergence Rate**). Let $P_{\text{true}}$ be the true underlying preference distribution and $P_{\text{learned}}$ be the policy trained on $n$ independent preferred samples per query drawn from $P_{\text{true}}$. Under standard assumptions (detailed in Appendix D), the expected Total Variation (TV) distance between the learned and true distributions decreases as:

$$\mathbb{E}[D_{TV}(P_{\text{learned}}, P_{\text{true}})] = \tilde{\mathcal{O}} \left( \frac{1}{\sqrt{n}} \right). \tag{9}$$

This theorem establishes that leveraging a larger set of $n$ preferred responses yields a provably tighter approximation of the ideal alignment policy than methods like DPO that rely on single-pair ($n = 1$) signals. A detailed proof is provided in Appendix E.

**2. Reliability of Set-Wise vs. Full-Ranking Methods.**   Theorem 1 establishes the benefit of using $n$ responses. This raises the question of *how* to best utilize them. We contrast two primary partitioning strategies for $n$ responses: (1) the $n$-**bin partition** (e.g., Plackett-Luce), which requires a full, strict ordering, and (2) the **2-bin partition** (MPO), which only requires a single partition into "accepted" and "rejected" sets. The reliability of each method depends on the integrity of its partitions under noisy reward signals. We analyze the probability of a correct partition for both methods under a standard additive Gaussian noise model.

**Theorem 2** (**Reliability under Spacing-Scaled Noise**). We analyze a formal noise model (see Appendix F) where the true score gaps scales as $\mathcal{O}(1/k)$. We denote the success probability of MPO as $\Pr\left(\text{MPO\_corr}\right)$ and for Plackett-Luce as $\Pr\left(\text{PL\_corr}\right)$. Under this *deterministic uniform spacing* model if the noise is additive Gaussian such that $\sigma \leq \beta/k$, then:

$$\Pr\left(\text{MPO\_corr}\right) \geq 1 - C(\beta) > 0, \quad \Pr\left(\text{PL\_corr}\right) \leq \exp(-\gamma(\beta)k), \quad \text{for } \gamma(\beta) > 0. \tag{10}$$

This analysis establishes MPO as a more robust method for learning from multiple, noisy preferences, avoiding the data-inefficiency of DPO while mitigating the rapid reliability degradation of full-ranking methods. The full proof is in Appendix F.

**3. Stationary Point of the MPO Loss.**   Finally, a crucial property of the MPO loss function (Eq. 5) concerns its optimization landscape. We find that at its stationary points, the model learns to assign vanishing probabilities to non-preferred responses. This differs from cross-entropy based approaches like InfoNCA, which match a target distribution. Detailed proof in Appendix H.

**Lemma 1** (**Vanishing Probability for Non-Preferred Responses**). For the MPO loss function, a stationary point is achieved as the model probabilities for all responses $y_i$ in the non-preferred set $\mathcal{Y}^-$ approach zero, i.e., $P_\theta(y_i \mid x) \rightarrow 0$ for all $y_i \in \mathcal{Y}^-$.

| Method | Mistral-Base (7B) | | | | Llama-3-Base (8B) | | | |
|---|---|---|---|---|---|---|---|---|
| | LC (%) | WR (%) | Arena-Hard | MT-Bench | LC (%) | WR (%) | Arena-Hard | MT-Bench |
| SFT[1] | 8.4 | 6.2 | 1.3 | 6.3 | 6.2 | 4.6 | 3.3 | 6.6 |
| KTO[1] | 13.1 | 9.1 | 5.6 | 6.8 | 14.2 | 12.4 | 12.5 | **7.8** |
| ORPO[1] | 14.7 | 12.2 | 7.0 | 7.0 | 12.2 | 10.6 | 10.8 | 7.6 |
| R-DPO[1] | 17.4 | 12.8 | 10.5 | 7.0 | 17.6 | 14.4 | 17.2 | 7.5 |
| GRPO | 10.1 | 6.7 | 6.3 | 6.8 | 8.2 | 5.7 | 6.0 | 7.1 |
| DPO | 16.6 | 13.8 | 12.7 | 6.7 | 16.9 | 14.1 | **18.5** | _7.7_ |
| NC2-DPO | 14.9 | 11.7 | 8.8 | 6.9 | 16.3 | 13.1 | 13.7 | 7.5 |
| InfoNCA | 14.8 | 10.8 | 9.7 | 7.0 | 15.9 | 12.9 | 14.8 | 7.6 |
| MPO 1vsk | 15.2 | 11.5 | 10.1 | 7.1 | 17.3 | 13.5 | 15.9 | 7.6 |
| MPO | _18.4_ | _14.4_ | **13.2** | 7.2 | 18.4 | 15.1 | _18.4_ | 7.5 |
| W-MPO | **20.3** | **14.9** | _12.8_ | _7.3_ | **20.1** | **15.6** | **18.5** | **7.8** |

Table 1: Comparison of methods on Mistral-Base and Llama-Base models in the off-policy setting with Ultrafeedback prompts and responses.

| Method | Mistral-Instruct (7B) | | | | Llama-3-Instruct (8B) | | | |
|---|---|---|---|---|---|---|---|---|
| | LC (%) | WR (%) | Arena-Hard | MT-Bench | LC (%) | WR (%) | Arena-Hard | MT-Bench |
| Base | 17.1 | 14.7 | 12.6 | 7.5 | 26.0 | 25.3 | 22.3 | 8.1 |
| DPO[1] | 26.8 | 24.9 | 16.3 | 7.6 | 40.3 | 37.9 | 32.6 | 8.0 |
| IPO[1] | 20.3 | 20.3 | 16.2 | **7.8** | 35.6 | 35.6 | 30.5 | **8.3** |
| KTO[1] | 24.5 | 23.6 | 17.9 | 7.7 | 33.1 | 31.8 | 26.4 | 8.2 |
| ORPO[1] | 24.5 | 24.9 | 20.8 | 7.7 | 28.5 | 27.4 | 25.8 | 8.0 |
| R-DPO[1] | 27.3 | 24.5 | 16.1 | 7.5 | 41.1 | 37.8 | 33.1 | 8.0 |
| MPO | **28.2** | **29.4** | **22.7** | 7.7 | **49.0** | **50.6** | **46.2** | 8.2 |

| Method | Qwen2.5-Instruct (14B) | | | | Qwen2.5-Instruct (32B) | | | |
|---|---|---|---|---|---|---|---|---|
| | LC (%) | WR (%) | Arena-Hard | MT-Bench | LC (%) | WR (%) | Arena-Hard | MT-Bench |
| Base | 36.4 | 30.8 | 68.3 | 8.9 | 40.9 | 33.6 | 71.9 | **9.2** |
| DPO | 60.3 | 63.8 | 84.3 | 8.9 | 59.0 | 63.6 | 83.8 | 9.0 |
| MPO | **62.3** | **65.9** | **86.8** | **9.0** | **65.8** | **66.9** | **87.9** | 9.1 |

Table 2: Results on Mistral-Instruct, Llama-Instruct and Qwen-Instruct models in the on-policy setting with Ultrafeedback prompts.

## 6 EXPERIMENTAL SETUP

We evaluate MPO across three training regimes: Offline, Online, and Iterative, using Mistral-7B and Llama-3-8B base models. Further details, including baselines, are in Appendix J.

**Off-policy Setting.** Following Tunstall et al. (2023), we first create supervised fine-tuned (SFT) models using UltraChat-200k (Ding et al., 2023). These SFT models initialize MPO and other methods for preference optimization on the UltraFeedback dataset (Cui et al., 2023).

**On-policy Setting.** We start with instruction-tuned models (Llama-3-8B-Instruct, Mistral-7B-Instruct). To align these models on policy, we generate 5 responses per UltraFeedback prompt (temperature 1.0) using the instruction-tuned models themselves, following Wu et al. (2024) and Meng et al. (2024). Responses are scored by Skywork-Reward-Llama-3.1-8B Liu et al. (2024b); the top-2 and bottom-2 are selected as preferred and rejected sets , discarding the median response.

**Iterative Setting** Using the same instruction-tuned base models, we adopt an iterative alignment framework akin to SPPO (Wu et al., 2024). The UltraFeedback prompt set is split for multi-round fine-tuning, with fresh responses (5 per prompt, temp 1.0, scored by Skywork-Reward-Llama-3.1-8B, top-2/bottom-2 selected) generated each round, following Wu et al. (2024).

We provide more details about our experimental setup in Appendix I and J.

---

[1] These are taken directly from the paper SIMPO: Simple Preference Optimization with a Reference-Free Reward (Meng et al., 2024)

| Method | Mistral-Instruct (7B) | | | | Llama-3-Instruct (8B) | | | |
|---|---|---|---|---|---|---|---|---|
| | LC (%) | WR (%) | Arena-Hard | MT-Bench | LC (%) | WR (%) | Arena-Hard | MT-Bench |
| SPPO[2] (Iteration 1) | 24.79 | 23.51 | 18.7 | 7.21 | 31.73 | 31.74 | 34.8 | 8.1 |
| MPO (Iteration 1) | **26.01** | **26.81** | **22.1** | **7.64** | **44.56** | **47.83** | **42.8** | **8.2** |
| SPPO[2] (Iteration 2) | 26.89 | 27.62 | 20.4 | 7.49 | 35.15 | 35.98 | 38.0 | 8.2 |
| MPO (Iteration 2) | **28.15** | **31.05** | **23.3** | **7.68** | **48.96** | **52.3** | **45.6** | **8.3** |
| SPPO[2] (Iteration 3) | 28.53 | 31.02 | 23.3 | 7.59 | 38.78 | 39.85 | 40.1 | 8.2 |
| MPO (Iteration 3) | **33.42** | **36.93** | **23.9** | **7.71** | **53.75** | **57.65** | **49.2** | **8.4** |

Table 3: Results on Mistral-Instruct and Llama-Instruct models in the iterative on-policy setting with Ultrafeedback prompts.

## 7 EXPERIMENTAL RESULTS

**SOTA Performance** We evaluate MPO against a broad spectrum of preference optimization baselines in three distinct training regimes: offline, online, and iterative. Across all settings, MPO consistently sets a new state-of-the-art. In the offline setup, as seen in Table 1, MPO outperforms established baselines such as R-DPO (Rafailov et al., 2024), InfoNCA (Chen et al., 2024a), and KTO (Ethayarajh et al., 2024). Notably, W-MPO achieves top scores.

In the online setup in Table 2, preference optimization is performed on instruction-tuned models. MPO again achieves SOTA performance across all benchmarks and model families. We also evaluate MPO in the iterative setting in Table 3. Compared against SPPO (Wu et al., 2024), the strongest prior iterative baseline, MPO yields significant improvements in every round.

**Performance with increasing number of responses**
Relying solely on a single best-worst pair per query misses out on rich signals available in settings with multiple responses per query. Specifically, models can produce diverse suboptimal responses, and learning from this can improve alignment. In Figure 4, we present results for `Llama-8B-Instruct` in the on-policy setting on AlpacaEval 2. We observe consistent gains as the number of responses per query increases from $n=2$ to $n=8$ under MPO, highlighting the benefits of using multiple responses during training.

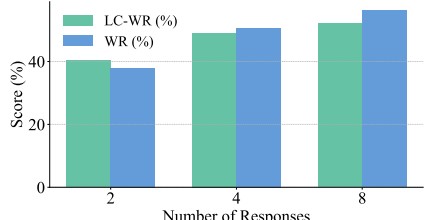

Figure 4: AlpacaEval2 performance with increasing number of model responses.

> *Key Takeaway: Including additional sub-optimal responses in* MPO *improves policy learning, challenging the convention that only top-ranked answers drive alignment performance.*

**Impact of deviation-based weighting in W-MPO** We evaluate the impact of W-MPO's core deviation-based weighting by comparing it to MPO. Table 4 underscores that deviation-aware weighting helps the policy learn from more separable supervision signals, leading to improved alignment.

| Method | Mistral (7B) | | Llama-3 (8B) | |
|---|---|---|---|---|
| | LC (%) | WR (%) | LC (%) | WR (%) |
| MPO | 28.2 | 29.4 | 49.0 | 50.6 |
| W-MPO | **29.5** | **30.6** | **52.1** | **52.5** |

Table 4: Comparison of MPO and W-MPO on AlpacaEval2 for Mistral and Llama Instruct models.

> *Key Takeaway:* MPO *supports the hypothesis that deviation based contrast amongst responses – not just their absolute reward – plays a critical role in preference optimization.*

**Ablations and Qualitative Examples in Appendix** In Appendix A, we provide additional experiments and qualitative examples (Appendix L). We show MPO maintains a superior training reward margin over baselines (Figure 8) and demonstrate its generality by extending DPO-variants like SimPO (Figure 9). We also provide hyperparameter ablations, finding MPO peaks at zero sampling temperature (Figure 11) and demonstrating that $\alpha$ and $\beta$ are critical for balancing convergence speed and stability (Figures 12, 13).

---

[2] These are taken directly from the paper SPPO: Self-Play Preference Optimization for Language Model Alignment (Wu et al., 2024)

**MPO achieves superior gains even with limited data** We examine how MPO performs under data-constrained settings for Mistral (7B), compared to DPO across varying proportions of training data. As shown in Figure 5, MPO outperforms DPO at varying percentages of training data. For instance, on LC-WR (%), MPO at 25% training data outperforms DPO at 50%. These improvements are a direct result of MPO's ability to leverage multiple responses per query, extracting more supervision signal.

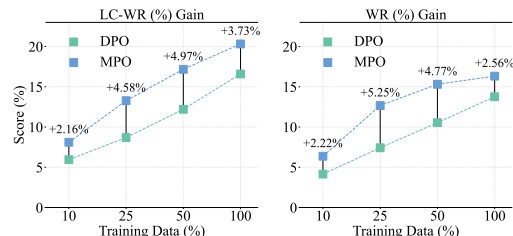

Figure 5: Comparison of MPO and DPO across varying training data sizes on AlpacaEval2.

*Key Takeaway:* *Even with limited training data,* MPO *consistently outperforms* DPO *by effectively utilizing multiple responses per query – highlighting the importance of multiple responses per query, over multiple queries, at a constant (query, response) budget.*

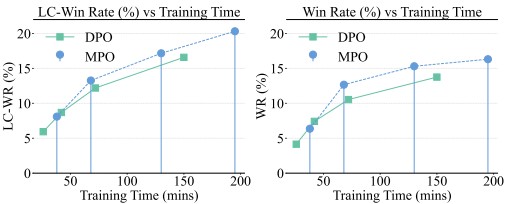

Figure 6: Comparison of MPO and DPO on AlpacaEval2 under fixed training-time budgets.

**Efficiency of MPO under fixed training budgets** While MPO naturally incurs longer per-step training time because it processes multiple responses per query, we investigate whether this cost translates into superior performance within a fixed wall-clock budget. Figure 6 shows that, despite the higher computation per batch, MPO consistently outperforms DPO at every time checkpoint, achieving substantial gains in both LC-WR and overall WR for Mistral (7B).

*Key Takeaway:* *Even under fixed training time,* MPO *outperforms* DPO*, showing its greater sample efficiency and stronger use of richer preference signals.*

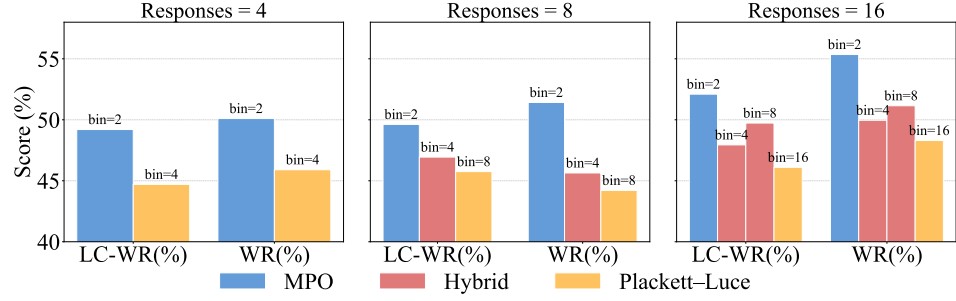

Figure 7: Performance comparison between MPO's 2-bin partitioning and Plackett-Luce's full ranking across varying numbers of responses per query for Llama-3-Instruct (8B) in online setting. MPO consistently outperforms the Plackett-Luce, with the gap widening as k increases

To empirically test Theorem 2, and compare the reliability of different partitioning granularities, we test MPO (2 bins) against a full-ranking Plackett-Luce ($k$ bins) and hybrid models with intermediate bins, varying the total number of responses $k \in \{4, 8, 16\}$ in Figure 7. MPO's 2-bin partition consistently outperforms all other configurations.The performance gap widens as $k$ increases.

*Key Takeaway:* *As formalized in Theorem 2, we find that* MPO*'s coarse-grained partitioning is a more effective learning mechanism than a full, fine-grained ranking, which results in better downstream performance. In fact, with increasing number of partitions, performance decreases.*

## 7.1 ISO-INFORMATION ANALYSIS

A key question to address is whether MPO's performance stems from its loss function design or simply from utilizing more data than standard DPO (which typically discards intermediate responses). To address this, we compare MPO against strict *Iso-Information* baselines that utilize the exact same set of $N$ responses and rewards:

- **Cross-Pairs DPO:** We construct every possible pair from the Cartesian product of the accepted ($S_+$) and rejected ($S_-$) sets and treat them as independent DPO updates. This tests if the volume of pairwise data explains MPO's gains.
- **Plackett-Luce & InfoNCA:** We also compare against the Plackett-Luce ranking model (which optimizes the full ranking probability) and InfoNCA (which uses a softmax-based contrastive loss over all $N$ responses).

As shown in Table 5, simply increasing the number of pairs via Cross-Pairs DPO actually degrades performance compared to the standard DPO baseline. Furthermore, MPO significantly outperforms both Plackett-Luce and InfoNCA. This confirms that MPO's advantage comes from the robustness of its set-wise binary contrast, rather than simply having access to the full response distribution.

| Method | Off-policy *(Mistral-7B)* | | On-policy *(Llama3-8B)* | |
|---|---|---|---|---|
| | LC-WR (%) | WR (%) | LC-WR (%) | WR (%) |
| DPO | 16.6 | 13.8 | 40.3 | 37.9 |
| Cross-pairs DPO (All Pairs) | 15.2 | 12.8 | 36.8 | 34.7 |
| Plackett-Luce | 13.7 | 9.6 | 44.6 | 45.5 |
| Info-NCA | 14.8 | 10.8 | 45.3 | 45.8 |
| **MPO (Ours)** | **20.3** | **14.9** | **49.0** | **50.6** |

Table 5: Comparison against Iso-Information Baselines

> *Key Takeaway: MPO's performance gains are intrinsic to it's novel set-wise objective, not merely a result of increased data.*

**Stability Across Reward Models** To validate that MPO policies generalize beyond the Skyworks RM, we trained MPO using three other models for reward generation: *GRM-Gemma-2B*, *ArmoRM-Llama3-8B*, and *PairRM*. As illustrated in Table 6, MPO yields consistent results regardless of the RM, confirming the robustness of MPO across RMs.

| Reward Model | LC-WR | WR | WR |
|---|---|---|---|
| Skywork-Reward-Llama-3.1-8B | 49.2 | 50.1 | 46.3 |
| GRM-Gemma-2B | **49.5** | **50.5** | **46.5** |
| ArmoRM-Llama3-8B-v0.1 | 48.5 | 49.5 | 45.5 |
| llm-blender/PairRM | 48.3 | 49.2 | 45.2 |

Table 6: Performance of MPO on Llama-3-8B-Instruct using reward scores from different RMs.

> *Key Takeaway: MPO's stability across diverse reward models confirms that the method is robust to RM-specific biases and offers generalized alignment improvements.*

## 8 DISCUSSION.

We introduced Multi-Preference Optimization (MPO), a novel post-training technique that generalizes DPO's contrastive loss to operate on entire sets of responses. This set-level contrastive formulation is grounded in our theoretical and empirical analysis. We first establish that leveraging more responses per query improves alignment, as shown by our convergence proof (Theorem 1) and empirical results (Figure 4). We then prove that MPO's 2-bin partition is fundamentally more robust to reward noise than competing full-ranking methods (Theorem 2), a finding confirmed empirically (Figure 7). Finally, MPO demonstrates state-of-the-art performance across off-policy, on-policy, and iterative settings (Tables 1, 2, 3), establishing a new, principled, and highly effective method for preference optimization.

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

# SUPPLEMENTARY MATERIALS

These supplementary materials provide additional details, derivations, and experimental results for our paper. The appendix is organized as follows:

- Section A presents additional experiments like hyperparameter ablations, and extensions of MPO.

- Section B presents a detailed related works section.

- Section C presents the core limitations of this work, and some avenues for future work.

- Section D presents a detailed overview of a variant of our main method which is weighted MPO.

- Section E presents a detailed proof of Theorem 1 using a Bernstein style error bound.

- Section F presents a detailed proof of Theorem 2 using a Gaussian tail bound.

- Section G provides a comprehensive comparison between the Group Contrastive Loss and InfoNCA Loss, including detailed gradient analyses.

- Section H offers a thorough characterization of stationary points for both the InfoNCA and Weighted Contrastive Loss functions.

- Section I describes the baselines used for comparison in our experimental evaluations, including various DPO implementations and alternative approaches.

- Section J describes the models, dataset and compute details for our experiments.

- Section K provides the implementation details of the reward loss computation, including the actual code used in our experiments.

- Section L provides examples of model responses using our model as compared with the base model.

- Section M provides comparative analysis of model responses using Gemini3 and GPT-5.

## A  ADDITIONAL EXPERIMENTS

### A.1  REWARD MARGIN ANALYSIS

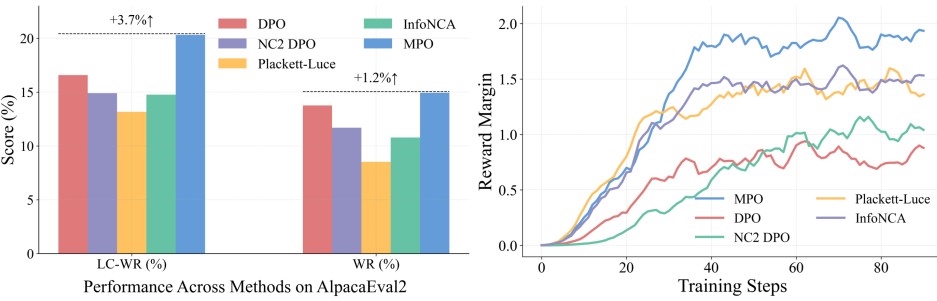

Figure 8: MPO's consistent reward margin advantage during training on Ultrafeedback enables clearer distinction between preferred and rejected responses, which translates into downstream evaluation as well on Mistral-7B.

We provide the reward margin, a training metric, as well as the downstream performance on AlpacaEval2 for these different related post-training optimization methods in Figure 8. The higher reward margin shows that the distinction between the top chosen and rejected response is higher in MPO vs. other baselines.

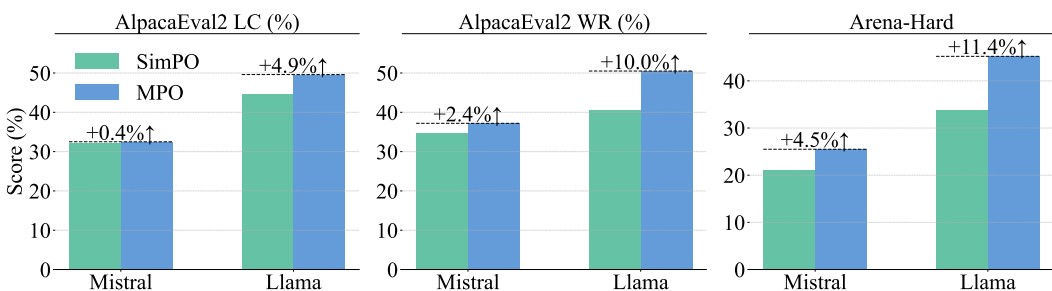

Figure 9: Evaluating MPO (Reference-Free) vs SIMPO on Mistral and Llama instruct models.

## A.2 UNIFYING DPO VARIANTS UNDER THE MPO SETTING

To demonstrate the generality, we explore how MPO can be instantiated on top of existing DPO-based methods, such as KTO (Ethayarajh et al., 2024), R-DPO (Park et al., 2024), and SimPO (Meng et al., 2024). These methods share the same pairwise contrastive structure as DPO, differing primarily in how they weigh or define the reward margin. Since MPO generalizes DPO to group-wise preference comparisons, it is naturally compatible with any loss that depends on contrast over preference data.

In Figure 9, we highlight a concrete instantiation: MPO (Reference-Free), where we extend SimPO's reward-based contrastive loss to operate in the multi-preference regime. This variant benefits from MPO's ability to better utilize responses. We observe consistent improvements across evaluation on downstream datasets. The results indicate that MPO (Reference-Free) leverages responses more effectively than SimPO.

## A.3 SENSITIVITY TO PARTITIONING STATISTICS

MPO partitions responses into Positive ($S_+$) and Negative ($S_-$) sets based on the mean reward. We compare this against **Median** and **Trimmed Mean** (removing top/bottom 10%).

Table 7 shows that while all statistics perform similarly in high-sample regimes ($N = 32$), the **Mean** is significantly more robust in low-sample regimes ($N = 4$). In sparse data settings, the Median often fails to capture the necessary margin for effective learning, leading to a performance drop. Thus, the Mean strikes the best balance between robustness and signal retention.

Table 7: Comparison of Mean, Median, and Trimmed Mean across high-resource ($N = 32$) and low-resource ($N = 4$) settings. The Mean statistic proves more robust when sample size is limited.

| Setting | Statistic | AlpacaEval LC-WR | AlpacaEval WR |
|---|---|---|---|
| **High-Sample** ($N = 32$) | **Mean (Ours)** | **52.1** | **52.5** |
| | Median | 51.8 | 51.4 |
| | Trimmed Mean | 51.7 | 51.6 |
| **Low-Sample** ($N = 4$, Offline) | **Mean (Ours)** | **20.1** | **15.6** |
| | Median | 17.8 | 14.7 |

## A.4 ROBUSTNESS TO REWARD NOISE INJECTION

To verify Theorem 2 empirically, we tested MPO's resilience to reward miscalibration. We injected Gaussian noise $\mathcal{N}(0, \sigma^2)$ into the ground-truth reward scores during the selection phase. As shown in Table 8, MPO maintains high performance even at $\sigma = 0.5$ and $\sigma = 1.0$, where pairwise ranking methods typically collapse. This validates that the coarse partitioning of MPO acts as an effective noise filter.

Table 8: Performance of Llama3-8B-Instruct on benchmarks with varying levels of Gaussian noise ($\sigma$) added to the reward signal

| Noise Level ($\sigma$) | AlpacaEval (LC-WR) | AlpacaEval (WR) | Arena-Hard |
|---|---|---|---|
| **0 (Baseline)** | **49.2** | **50.1** | **46.3** |
| 0.5 | 48.1 | 48.9 | 44.8 |
| 1.0 | 45.6 | 46.5 | 43.6 |
| 2.0 | 42.3 | 43.1 | 41.5 |

## A.5 ANALYSIS OF REWARD SPACING

To validate the practical validation of our theoretical noise analysis (Theorem 2), we analyzed the reward distribution of responses generated by Llama3-8B-Instruct.

Figure 10 illustrates the average reward spacing ($\Delta$) between adjacently ranked responses as the number of sampled responses ($k$) increases. We observe that the spacing shrinks drastically, following an approximate $O(1/k)$ trend.

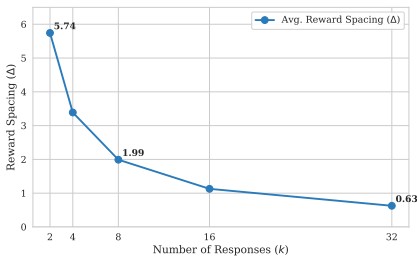

Figure 10: Decay of Reward Spacing ($\Delta$)

## A.6 HYPERPARAMETER ABLATIONS

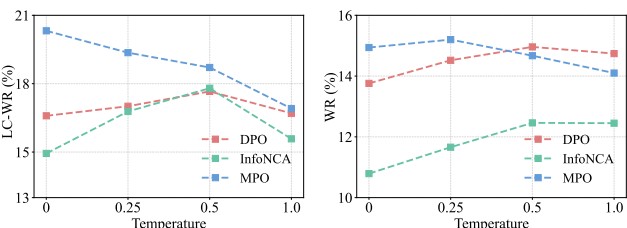

Figure 11: Performance variation of MPO for Mistral-Base with different sampling temperature on AlpacaEval2.

**Optimizing for Lower Temperatures** Fig. 11 evaluates MPO, DPO, and InfoNCA on AlpacaEval 2 (Mistral-Base 7B) against sampling temperature in the off-policy setting for Mistral-7B. MPO achieves its peak LC-WR of $\sim 20.5\%$ at zero temperature (greedy decoding), declining thereafter. Notably, MPO consistently leads in LC-Win Rate across all temperatures.

**Effect of $\beta$ on Optimization Dynamics.** In Figure 12, we conduct an ablation study to investigate the role of the temperature parameter $\beta$ in our optimization objective. As $\beta$ increases, the gradient magnitudes decrease due to the flattening of the sigmoid-like objective, effectively pushing the scaled reward term $\beta \cdot \log \frac{\pi_{policy}(y|x)}{\pi_{ref}(y|x)}$ outside the region where the derivative is steepest. This leads to slower updates and consequently a more conservative learning trajectory. In contrast, smaller values of $\beta$ keep the optimization in the high-sensitivity regime of the sigmoid, allowing for faster convergence. Empirically, we observe that using a smaller $\beta$ (e.g., $\beta = 0.01$) enables the model

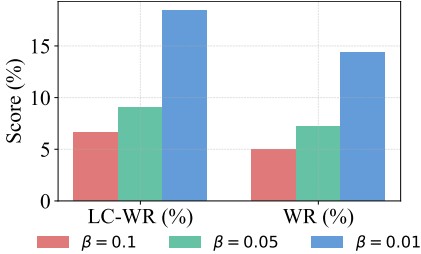

Figure 12: Performance variation of MPO for Mistral-Base with different $\beta$ values on AlpacaEval2.

to reach comparable performance to higher $\beta$ settings with significantly fewer epochs and smaller learning rates. This demonstrates that careful tuning of $\beta$ can not only preserve final accuracy but also substantially reduce training cost.

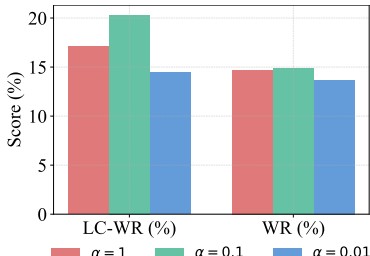

Figure 13: Performance variation of W-MPO for Mistral-Base with different $\alpha$ values on AlpacaEval2.

**Effect of $\alpha$ on Optimization Dynamics.**    In Figure 13, we analyze the impact of the reward normalization coefficient $\alpha$, which scales the raw reward signal before computing the deviation weight in W-MPO. This parameter plays a crucial role in balancing the sensitivity of the weighting function. A larger $\alpha$ (e.g., $\alpha = 1$) retains the raw reward's expressivity, but can lead to high variance in gradient updates, potentially destabilizing training. On the other hand, a very small $\alpha$ (e.g., $\alpha = 0.01$) overly compresses the reward differences, weakening the contrast between preferred and dispreferred responses—this leads to underutilized supervision and poor alignment, as evidenced by the drop in LC-WR from 20.3 to 14.43. We find that an intermediate setting (e.g., $\alpha = 0.1$) strikes a favorable balance, providing stable optimization while preserving informative contrast. This setting achieves the highest LC-WR and WR performance, demonstrating the importance of calibrating $\alpha$ to match the scale and distribution of reward scores.

## B    RELATED WORK

We will start this literature survey with a high level overview of the RLHF literature and then going deeper into the area of preference, and then multi-preference optimization relevant to our work.

**Broader RLHF Literature:**  Reinforcement Learning through Human feedback (RLHF) has emerged as a robust alignment algorithm for language models. The area broadly started of with works like Trust Region Policy Optimization (TRPO), and Proximal Policy Optimization (PPO) (Schulman, 2015; Schulman et al., 2017) which extend direct RL based methods by constraining the update space to within a trusted region and clipping policy updates to prevent instability respectively. Building upon earlier policy gradient methods (Sutton et al., 1999), PPO has been successfully applied to alignment tasks in Reinforcement Learning from Human Feedback (RLHF), allowing language models to produce outputs aligned with human preferences (Ziegler et al., 2019; Ouyang et al., 2022). Its simplicity and efficiency make it a standard approach for fine-tuning large-scale models. Prior to PPO, Trust Region Policy Optimization (TRPO) (Schulman, 2015) introduced constraints to improve learning stability, influencing the development of PPO. Early applications of policy gra-

dient methods in natural language processing (Ranzato et al., 2015) demonstrated the potential of reinforcement learning for language model training.

**Preference Optimization:** Direct Preference Optimization (DPO) simplifies the alignment of language models by optimizing a contrastive loss directly over paired preference data, bypassing the intermediate step of reward modeling (Rafailov et al., 2024). Unlike RLHF, DPO does not require explicit reward functions, making it computationally efficient and suitable for limited preference datasets. Recent extensions of DPO, such as Identity Preference Optimization (IPO) (Azar et al., 2024), self-play preference optimization (Wu et al., 2024), preference ranking optimization (Song et al., 2024), rejection sampling optimization (Liu et al., 2023), and generalized preference optimization (Tang et al., 2024) are amongst the other recent works improve on the DPO method.

Beyond the foundational pairwise approaches and their direct extensions, numerous recent works have proposed methods that adapt or refine DPO-like strategies, often eliminating the need for separate reward modeling or reference models.

**Alternative Approaches Without Full Reward Modeling.** Ethayarajh et al. (2024) propose **KTO**, a framework inspired by prospect theory that directly learns whether a response is globally desirable or undesirable, thereby removing the requirement of having multiple positive examples per instruction. Zeng et al. (2024) focus on token-level alignment in **TDPO**, imposing forward KL divergence constraints for each token rather than solely for the final output. This fine-grained approach can mitigate the mode-collapse issues sometimes observed in sequence-level alignment. Meanwhile, Dong et al. (2023) introduce a list-wise method called **RAFT**, where the model fine-tunes on the best response from each sampled set of $k$ candidates, iteratively converging toward an optimal subset policy. By contrast, Yuan et al. (2023) center on rank-based supervision through **RRHF**, which combines a rank loss with standard supervised signals to ensure the model maintains stronger probabilities on higher-ranked (i.e., better) responses and less on suboptimal responses.

**Enhancing DPO with Additional Objectives and Training Schemes.** Other works further modify or reinterpret the DPO loss to incorporate new constraints or to remove the need for a reference model. Chen et al. (2024c) propose **SPIN**, which treats the model as part of a two-player adversarial game, obviating separate reward modeling by training with a discriminator that distinguishes human from machine responses. **CPO** (Xu et al., 2024) reworks the DPO objective by removing the reference-model term and adding a behavior cloning regularizer. Similarly, **ORPO** (Hong et al., 2024) folds preference optimization into a negative log-likelihood objective via an odds-ratio penalty, thereby unifying supervised fine-tuning (SFT) and preference training. In **SimPO**, Meng et al. (2024) remove the reference model and incorporate a length normalization to address verbosity issues that can skew preference data. Likewise, **R-DPO** (Park et al., 2024) and **LD-DPO** (Liu et al., 2024c) specifically tackle length bias by injecting additional regularizers or by explicitly separating length-based preferences from other factors. For instance, LD-DPO modifies the training set to handle length constraints, preventing performance drops on standard benchmarks while mitigating length exploitation in preference tasks.

**Refining Training Regimens for Preference Data.** A final family of works emphasizes how training procedures or data usage can be systematically improved. For instance, Kim et al. (2024) propose **sDPO**, a step-wise learning method partitioning preference data to stabilize training. **IRPO** (Pang et al., 2024) enhances chain-of-thought reasoning by incorporating a negative log-likelihood term for the chosen solution path, thus nudging LLMs toward robust multi-step reasoning. **OFS-DPO** (Qi et al., 2024) trains two LoRA modules at different paces—one faster, one slower—to sustain gradient momentum and to adapt more efficiently. Lastly, Yuan et al. (2024) tackle verbosity with **LIFT-DPO**, an approach that augments preference data with length-control instructions, ensuring that the model does not exploit response length to inflate its preference scores.

**Multi-Preference Optimization:** Traditional preference optimization methods, like DPO, consider pairwise comparisons. However, datasets such as UltraFeedback (Cui et al., 2023) highlight the necessity of multi-preference optimization. Multi-preference methods, such as InfoNCA (Chen et al., 2024a), leverage all available positive and negative responses simultaneously, reducing alignment bias and better approximating the true preference distribution. These methods mitigate limitations inherent to pairwise approaches by incorporating the diversity of acceptable and suboptimal re-

sponses. Earlier works in search have also used multiple user preferences to optimize models in various applications such as search (Joachims, 2002).

**Reward Modeling in Preferences:** Reward modeling is essential for translating qualitative human feedback into quantitative metrics that guide AI behavior optimization. Traditional methods, such as Reinforcement Learning from Human Feedback (RLHF), utilize reward models trained on human annotations to inform policy updates (Christiano et al., 2017; Stiennon et al., 2020). Early approaches like inverse reinforcement learning (Ng et al., 2000) and apprenticeship learning (Abbeel & Ng, 2004) demonstrated the feasibility of inferring reward functions from observed behaviors. Recent advancements have diversified reward modeling techniques. For instance, the Adversarial Preference Optimization (APO) framework employs adversarial training to adapt reward models to the evolving generation distribution of language models (Cheng et al., 2023).

**Noise Contrastive Estimation and InfoNCA:** Contrastive learning, particularly methods like InfoNCE (Oord et al., 2018), maximizes mutual information between positive samples while discriminating against negatives. In the language domain, Klein & Nabi (2024) leverage a perplexity-based contrastive objective to reduce toxic language generation while preserving the model's overall utility. InfoNCA adapts these principles for preference optimization, aligning responses with scalar rewards through noise-contrastive estimation (Chen et al., 2024a). Despite its strengths, InfoNCA can overemphasize less informative negative samples, which motivates methods like Mpo that dynamically weigh responses based on deviation from the mean reward.

**UltraFeedback Dataset:** The UltraFeedback dataset (Cui et al., 2023) is a significant advancement in preference-based training resources. It comprises GPT-4 annotated feedback for over 64,000 instructions, including scalar reward evaluations. UltraFeedback has been pivotal in developing models like UltraLM-13B-PPO and UltraRM, which achieve state-of-the-art performance across benchmarks such as AlpacaEval. This dataset's granularity enables advanced preference optimization methods like Mpo to leverage diverse response quality levels effectively.

## C  LIMITATIONS

We note the following limitations and scope considerations for the present work.

**Dependence on Upstream Reward Signals**   Like all methods in the DPO-family of preference optimization, Mpo's efficacy is linked to the quality of the preference data from which we generate the preference sets. Our work does not explore the generation of these reward signals; rather, it is focused on a novel, robust mechanism for *learning from* them once they are provided. While our theoretical (Theorem 2) and empirical (Figure 7) results demonstrate MPO's superior robustness to noise compared to full-ranking methods, extreme miscalibration of the reward source remains a systemic challenge for the entire field.

**Scope of Preference Model**   Our method generalizes DPO from a pairwise ($1 \, vs \, 1$) contrast to a set-wise ($n \, vs \, m$) contrast. This is a deliberate design choice that models a single "accepted" vs. "rejected" partition, which we prove is highly robust. We acknowledge that human preferences can be more granular, such as in a complete $k$-way ranking. Our work does not address these finer-grained preference structures, as our focus is on creating a scalable and noise-resistant alternative to pairwise DPO, not on replicating a full ranking.

**Computational Scope**   The Mpo loss computation scales linearly with the total number of responses $k$ per prompt. Our experiments and analysis focus on $k \leq 16$, which is a common regime in practical alignment pipelines. The exploration of MPO's computational performance and scaling properties in a "large-$k$" regime (e.g., $k > 100$) is a distinct research problem that falls outside the scope of this paper.

## LLM USAGE STATEMENT

The authors acknowledge the use of a large language model (LLM) in the preparation of this manuscript. The LLM was utilized as a collaborative writing assistant for editing and refining the

text for clarity, grammar, and conciseness. Additionally, the LLM assisted in generating Python code used for data visualization in several of the paper's figures. All core intellectual contributions, including the theoretical analysis, experimental design, and interpretation of results, were conducted by the human authors.

# D    DETAILED FORMULATION OF WEIGHTED MPO (W-MPO)

While the core Multi-Preference Optimization (MPO) framework, as presented in Section 4, effectively leverages sets of preferred ($\mathcal{Y}_x^+$) and dispreferred ($\mathcal{Y}_x^-$) responses, it treats all responses within each set with uniform importance. However, many modern alignment pipelines, particularly those employing on-policy or iterative generation and evaluation, utilize powerful reward models (RMs) that provide fine-grained scalar quality scores $S(y)$ for each generated response $y$. These scores often reveal significant variance in quality even within the pre-defined preferred or dispreferred sets. Weighted MPO (W-MPO) is designed to exploit this richer, quantitative feedback.

## D.1    MOTIVATION FOR WEIGHTED MPO

In contemporary LLM alignment, especially in on-policy settings (e.g., RLHF with a learned reward model, or iterative DPO-like methods where a model generates responses that are then scored), obtaining exhaustive human pairwise preferences for all generated outputs can be prohibitively expensive or slow. Instead, an automated reward model $RM(y \mid x)$ is often used to assign a scalar quality score $S(y)$ to each response $y$ generated for a prompt $x$. These scalar scores $S(y)$ offer several advantages:

1. **Granularity:** They provide a more nuanced measure of quality than simple binary (chosen/rejected) or pairwise preferences. A response $y_1$ might be preferred over $y_2$, but $S(y_1)$ can also indicate *how much* better it is, or how it compares to a global quality scale.

2. **Efficiency:** RMs can score a large volume of responses quickly, enabling larger-scale and more frequent preference data collection, crucial for iterative alignment.

3. **Handling Multiple Responses:** When multiple ($N_x > 2$) responses are generated per prompt (e.g., via diverse beam search, multiple sampling seeds, or from different model variants), RMs can score all of them, providing a rich landscape of quality.

The core MPO method, by operating on sets $\mathcal{Y}_x^+$ and $\mathcal{Y}_x^-$ (which might themselves be derived from $S(y)$ via a threshold like $S_{\text{mean}}(x)$), does not fully capitalize on the information that some responses in $\mathcal{Y}_x^+$ are "more preferred" (higher $S(y)$) than others, or that some in $\mathcal{Y}_x^-$ are "more dispreferred" (lower $S(y)$). Responses that are significant outliers in terms of their quality scores (i.e., those deviating most from an average or baseline quality) are often the most informative for training. For instance, an exceptionally good response provides a strong positive signal, while a particularly problematic response offers a clear negative signal to learn from.

W-MPO aims to incorporate this finer-grained information by weighting the contribution of each response $y$ in the MPO loss based on its scalar score $S(y)$. Specifically, it uses the deviation of $S(y)$ from a reference point (e.g., the mean score $S_{\text{mean}}(x)$ for the current prompt) to modulate the response's impact. This allows W-MPO to:

- **Prioritize Informative Samples:** Give more importance to responses that are either exceptionally good or exceptionally bad, as these are strong learning signals.

- **Implement a Data-Driven Curriculum:** Naturally focus the model's attention on responses from which it can learn the most about the boundaries of desired behavior.

- **Enhance Stability and Robustness:** By considering the full spectrum of quality scores, W-MPO can potentially lead to more stable and robust alignment compared to methods that only consider coarse preference signals.

This approach is particularly relevant when $S(y)$ comes directly from a reward model, as it allows the alignment process to more faithfully reflect the nuances captured by that RM.

## D.2 W-MPO FORMULATION

Let $\mathcal{D}_{\text{W-MPO}}$ be a dataset where each instance consists of a query $x$, a set of $N_x$ responses $\mathcal{Y}_x = \{y_1, \ldots, y_{N_x}\}$ for that query, and their corresponding scalar quality scores $\{S(y_1), \ldots, S(y_{N_x})\}$.

**Response Partitioning and Deviation Calculation.** First, for each query $x$, we calculate the mean quality score:

$$S_{\text{mean}}(x) = \frac{1}{N_x} \sum_{j=1}^{N_x} S(y_j). \tag{11}$$

The responses are then partitioned into preferred and dispreferred sets:

$$\mathcal{Y}_x^+ = \{y \in \mathcal{Y}_x \mid S(y) > S_{\text{mean}}(x)\}, \tag{12}$$

$$\mathcal{Y}_x^- = \{y \in \mathcal{Y}_x \mid S(y) \le S_{\text{mean}}(x)\}. \tag{13}$$

If $\mathcal{Y}_x^+$ is empty, the sample for query $x$ is typically skipped.

Next, we define a signed *deviation term* $\Delta W(y)$ for each response $y \in \mathcal{Y}_x$:

$$\Delta W(y) = S(y) - S_{\text{mean}}(x). \tag{14}$$

This term, $\Delta W(y)$, is positive for $y \in \mathcal{Y}_x^+$ and non-positive for $y \in \mathcal{Y}_x^-$. It quantifies how much better or worse a response is compared to the average quality for that query.

**Additive Logit Adjustment.** W-MPO incorporates this deviation $\Delta W(y)$ by additively adjusting the logit (scaled implicit preference score) of each response. The base implicit score is $r_\theta(y \mid x) = \log(\pi_\theta(y \mid x)/\pi_{\text{ref}}(y \mid x))$, as defined in Eq. 3. The modified logit, $r'_\theta(y \mid x)$, used in W-MPO is:

$$r'_\theta(y \mid x; \alpha_w) = \beta \cdot r_\theta(y \mid x) + \alpha_w \cdot \Delta W(y), \tag{15}$$

where $\beta$ is the inverse temperature scaling the base DPO-like score, and $\alpha_w$ is a hyperparameter controlling the magnitude of the deviation-based adjustment. A positive $\alpha_w$ will increase the effective logit for responses with $S(y) > S_{\text{mean}}(x)$ and decrease it for responses with $S(y) < S_{\text{mean}}(x)$, with the magnitude of adjustment proportional to $|\Delta W(y)|$.

**W-MPO Loss Function.** The W-MPO loss function retains the structure of the core MPO loss (Eq. 5) but utilizes these modified logits $r'_\theta(y \mid x; \alpha_w)$:

$$\mathcal{L}_{\text{W-MPO}}(\theta) = -\mathbb{E}_{(x, \mathcal{Y}_x, \{S(y)\}) \sim \mathcal{D}_{\text{W-MPO}}} \left[ \log \left( \frac{\sum_{y \in \mathcal{Y}_x^+} \exp(r'_\theta(y \mid x; \alpha_w))}{\sum_{y' \in \mathcal{Y}_x^+ \cup \mathcal{Y}_x^-} \exp(r'_\theta(y' \mid x; \alpha_w))} \right) \right]. \tag{16}$$

This additive adjustment in the logit space (Eq. 15) is equivalent to applying a multiplicative exponential weight $w_{\exp}(y) = \exp(\alpha_w \cdot \Delta W(y))$ to the original base strength term $\exp(\beta \cdot r_\theta(y \mid x))$ within the sums of the MPO loss. The W-MPO objective thus encourages the policy $\pi_\theta$ to more strongly prefer/disprefer responses that are further above/below the mean quality score, respectively.

## D.3 ALGORITHM FOR W-MPO

The detailed training procedure for W-MPO is presented in Algorithm 2.

**Algorithm 2** Detailed Weighted Multi-Preference Optimization (W-MPO)

---

1: **Input:** Initial policy parameters $\theta_0$; dataset $\mathcal{D}_{\text{W-MPO}}$, where each sample contains $(x, \mathcal{Y}_x, \{S(y)\}_{y \in \mathcal{Y}_x})$; reference model $\pi_{\text{ref}}$; inverse temperature $\beta$; W-MPO weight scaling hyperparameter $\alpha_w$; learning rate $\eta$; number of training iterations/epochs $T$.

2: **Output:** Optimized model parameters $\theta_T$.

3: Initialize policy model parameters $\theta \leftarrow \theta_0$.

4: **for** $t = 1$ **to** $T$ **do**

5:      Initialize total loss for epoch/iteration $\mathcal{L}_{\text{epoch}} \leftarrow 0$.

6:      **for all** sample $(x, \mathcal{Y}_x, \{S(y)\}_{y \in \mathcal{Y}_x})$ in $\mathcal{D}_{\text{W-MPO}}$ (typically processed in mini-batches) **do**

7:          Compute $S_{\text{mean}}(x) \leftarrow \frac{1}{|\mathcal{Y}_x|} \sum_{y_j \in \mathcal{Y}_x} S(y_j)$. (Ref. Eq. 11)

8:          Define $\mathcal{Y}_x^+ \leftarrow \{y \in \mathcal{Y}_x \mid S(y) > S_{\text{mean}}(x)\}$. (Ref. Eq. 12)

9:          Define $\mathcal{Y}_x^- \leftarrow \{y \in \mathcal{Y}_x \mid S(y) \leq S_{\text{mean}}(x)\}$. (Ref. Eq. 13)

10:         **if** $|\mathcal{Y}_x^+| = 0$ **then**

11:             **continue** {Skip sample if no preferred responses after partitioning}

12:         **end if**

13:         Let $\mathcal{Y}_{\text{all}} = \mathcal{Y}_x^+ \cup \mathcal{Y}_x^-$.

14:         Create a map $R'_\theta$ to store modified logits.

15:         **for** $y \in \mathcal{Y}_{\text{all}}$ **do**

16:             Compute base score: $r_\theta(y \mid x) \leftarrow \log\left(\frac{\pi_\theta(y|x)}{\pi_{\text{ref}}(y|x)}\right)$. (Ref. Eq. 3)

17:             Compute deviation term: $\Delta W(y) \leftarrow S(y) - S_{\text{mean}}(x)$. (Ref. Eq. 14)

18:             Compute modified logit: $r'_\theta(y \mid x) \leftarrow \beta \cdot r_\theta(y \mid x) + \alpha_w \cdot \Delta W(y)$. (Ref. Eq. 15)

19:             $R'_\theta[y] \leftarrow r'_\theta(y \mid x)$.

20:         **end for**

21:         Numerator term $N_x \leftarrow \sum_{y \in \mathcal{Y}_x^+} \exp(R'_\theta[y])$.

22:         Denominator term $D_x \leftarrow \sum_{y' \in \mathcal{Y}_{\text{all}}} \exp(R'_\theta[y'])$.

23:         Sample loss $L_x \leftarrow -\log(N_x/D_x)$, ensuring $N_x/D_x \in (0, 1]$. (Ref. Eq. 16)

24:         Accumulate loss (e.g., $\mathcal{L}_{\text{epoch}} \leftarrow \mathcal{L}_{\text{epoch}} + L_x$).

25:      **end for**

26:      Compute gradient $\nabla_\theta \mathcal{L}_{\text{epoch}}$ (typically averaged over batch size).

27:      Update model parameters: $\theta \leftarrow \theta - \eta \nabla_\theta \mathcal{L}_{\text{epoch}}$.

28: **end for**

29: **return** $\theta$.

---

# E    PROOF OF THEOREM 1: TV CONVERGENCE RATE IN LOW-DIMENSIONAL RESPONSE MODELS

This appendix provides the formal setup, assumptions, and proof for Theorem 1 in the setting where the response space may be infinite (or very large), but the true distribution lies in a $k$-dimensional parametric family. Under the stated regularity conditions, the learned preference distribution converges to the true distribution in total variation at the rate $\tilde{O}(\sqrt{k/n})$. For fixed $k$, this is the canonical $\tilde{O}(n^{-1/2})$ rate.

## E.1    FORMAL SETUP AND ASSUMPTIONS

We fix a single query $x$ and its response space $\mathcal{Y}(x)$, which need not be finite. Let $(\mathcal{Y}(x), \mathcal{A}, \mu)$ be a measurable space endowed with a reference $\sigma$-finite measure $\mu$ (counting measure in the discrete case, Lebesgue otherwise). We consider a $k$-dimensional exponential-family model for the conditional density of responses given $x$:

$$p_\theta(y \mid x) \;=\; \exp\left\{\theta^\top \phi(y) - A(\theta)\right\} \quad \text{with} \quad A(\theta) \;=\; \log \int_{\mathcal{Y}(x)} \exp\left(\theta^\top \phi(z)\right) d\mu(z),$$

where $\phi : \mathcal{Y}(x) \to \mathbb{R}^k$ is a fixed feature map.

**A1. True Distribution (Well-Specified Exponential Family).** There exists $\theta_\star \in \Theta \subset \mathbb{R}^k$ such that the true preference distribution admits density $p_{\text{true}}(\cdot \mid x) = p_{\theta_\star}(\cdot \mid x)$ with respect to $\mu$.

**A2. Feature Regularity.** The feature map is uniformly bounded: $\|\phi(y)\|_2 \leq R$ for $\mu$-a.e. $y$, for some finite $R > 0$.

**A3. Strong Convexity at $\theta_\star$.** The log-partition function $A(\theta)$ is twice differentiable on a neighborhood of $\theta_\star$, with Hessian $\nabla^2 A(\theta_\star) = \text{Cov}_{p_{\theta_\star}}[\phi(Y)]$ satisfying a spectral lower bound

$$\lambda_{\min}\left(\nabla^2 A(\theta_\star)\right) \;\geq\; m_0 > 0.$$

(This is an identifiability and local curvature condition.)

**A4. Observation Model (i.i.d. Responses).** We observe $n$ i.i.d. responses $Y_{1:n} \sim p_{\theta_\star}(\cdot \mid x)$.

**A5. Learner (MLE in the Model Class).** The learned policy $p_{\text{learn}}(\cdot \mid x) = p_{\hat{\theta}_n}(\cdot \mid x)$ is the maximum-likelihood estimator:

$$\hat{\theta}_n \in \arg\max_{\theta \in \Theta} \frac{1}{n} \sum_{i=1}^n \log p_\theta(Y_i \mid x) \quad \left(\text{equivalently, } \hat{\theta}_n \in \arg\min_{\theta \in \Theta} \widehat{A}_n(\theta) - \theta^\top \widehat{\mu}_n\right),$$

where $\widehat{\mu}_n := \frac{1}{n} \sum_{i=1}^n \phi(Y_i)$ and $\widehat{A}_n(\theta) := A(\theta)$.

Our target metric is the total variation (TV) distance:

$$\text{TV}\left(p_{\text{learn}}(\cdot \mid x), p_{\text{true}}(\cdot \mid x)\right) := \frac{1}{2} \int_{\mathcal{Y}(x)} \left|p_{\hat{\theta}_n}(y \mid x) - p_{\theta_\star}(y \mid x)\right| d\mu(y).$$

## E.2    PROOF INGREDIENTS

We collect two standard components: a finite-sample parameter error bound for the MLE and a smoothness-based link from parameter error to TV error.

**Lemma 2** (Finite-sample parameter error). Under **A1**–**A4**, for all $\delta \in (0, 1)$, with probability at least $1 - \delta$,

$$\|\hat{\theta}_n - \theta_\star\|_2 \;\leq\; \frac{2}{m_0} \left\|\widehat{\mu}_n - \mu_\star\right\|_2 \;\leq\; \frac{C_1 R}{m_0} \sqrt{\frac{k + \log(1/\delta)}{n}},$$

where $\mu_\star := \mathbb{E}_{p_{\theta_\star}}[\phi(Y)]$ and $C_1 > 0$ is a universal constant.

*Proof sketch.* By first-order optimality, $\nabla A(\hat{\theta}_n) = \widehat{\mu}_n$ and $\nabla A(\theta_\star) = \mu_\star$. By the mean-value expansion and strong convexity at $\theta_\star$, there exists $\tilde{\theta}$ on the segment connecting $\theta_\star$ and $\hat{\theta}_n$ such that

$$\widehat{\mu}_n - \mu_\star = \nabla A(\hat{\theta}_n) - \nabla A(\theta_\star) = \nabla^2 A(\tilde{\theta})\,(\hat{\theta}_n - \theta_\star), \quad \Rightarrow \quad \|\hat{\theta}_n - \theta_\star\|_2 \leq \frac{\|\widehat{\mu}_n - \mu_\star\|_2}{\lambda_{\min}(\nabla^2 A(\tilde{\theta}))}.$$

Continuity of the Hessian and **A3** give $\lambda_{\min}(\nabla^2 A(\tilde{\theta})) \geq m_0/2$ on a small neighborhood (absorbed into constants). The concentration bound $\|\widehat{\mu}_n - \mu_\star\|_2 \leq C_1 R \sqrt{(k + \log(1/\delta))/n}$ follows by vector-valued Hoeffding or Bernstein using $\|\phi(Y)\|_2 \leq R$. $\square$

**Lemma 3** (From parameter error to TV via smoothness)**.** Suppose $A$ is twice differentiable on a neighborhood of $\theta_\star$ with Hessian upper bounded as $\nabla^2 A(\theta) \preceq L_0 I_k$ on that neighborhood. Then

$$D_{\mathrm{KL}}(p_{\theta_\star} \,\|\, p_{\hat{\theta}_n}) \;\leq\; \frac{L_0}{2}\,\|\hat{\theta}_n - \theta_\star\|_2^2, \qquad \mathrm{TV}(p_{\hat{\theta}_n}, p_{\theta_\star}) \;\leq\; \sqrt{\frac{1}{2}\,D_{\mathrm{KL}}(p_{\theta_\star} \,\|\, p_{\hat{\theta}_n})} \;\leq\; \sqrt{\frac{L_0}{4}}\,\|\hat{\theta}_n - \theta_\star\|_2.$$

*Proof.* For exponential families, $D_{\mathrm{KL}}(p_{\theta_\star}\|p_{\hat{\theta}_n}) = A(\hat{\theta}_n) - A(\theta_\star) - (\hat{\theta}_n - \theta_\star)^\top \nabla A(\theta_\star)$. By the mean-value form of Taylor's theorem and the Hessian bound, $D_{\mathrm{KL}} \leq \frac{1}{2}(\hat{\theta}_n - \theta_\star)^\top \nabla^2 A(\bar{\theta})(\hat{\theta}_n - \theta_\star) \leq \frac{L_0}{2}\|\hat{\theta}_n - \theta_\star\|_2^2$. Pinsker's inequality yields the TV bound. $\square$

### E.3 THEOREM AND PROOF

**Theorem 3** (**Convergence Rate in TV for Low-Dimensional Models**)**.** Under **A1–A5**, and assuming the local Hessian bounds

$$m_0 I_k \;\preceq\; \nabla^2 A(\theta) \;\preceq\; L_0 I_k \quad \text{for all } \theta \text{ in a neighborhood of } \theta_\star,$$

the learned distribution satisfies, for any $\delta \in (0,1)$, with probability at least $1 - \delta$,

$$\mathrm{TV}(p_{\mathrm{learn}}(\cdot \mid x), p_{\mathrm{true}}(\cdot \mid x)) \;\leq\; \frac{\sqrt{L_0}\,C_1 R}{m_0}\sqrt{\frac{k + \log(1/\delta)}{n}} \;=\; \tilde{\mathcal{O}}\left(\sqrt{\frac{k}{n}}\right). \tag{17}$$

*Proof.* Combining Lemma 2 with Lemma 3 gives

$$\mathrm{TV}(p_{\hat{\theta}_n}, p_{\theta_\star}) \;\leq\; \sqrt{\frac{L_0}{4}}\,\|\hat{\theta}_n - \theta_\star\|_2 \;\leq\; \sqrt{\frac{L_0}{4}} \cdot \frac{C_1 R}{m_0}\sqrt{\frac{k + \log(1/\delta)}{n}}.$$

Absorb constants to obtain the displayed bound. $\square$

*Remark* 1 (Fixed-$k$ specialization and relation to finite-support bounds)**.** When $k$ is treated as a constant independent of $n$, the bound reduces to $\tilde{O}(n^{-1/2})$. In the special case of a finite response set of size $m$ and the nonparametric MLE on the simplex, one can take $k = m - 1$, $R \leq 2$, and recover the classical $\tilde{O}(\sqrt{m/n})$ dependence, consistent with multinomial concentration.

*Remark* 2 (Interpretation of $m_0$ and $L_0$)**.** The constants $m_0$ and $L_0$ are, respectively, lower and upper spectral bounds on the Fisher information $\nabla^2 A(\theta)$ in a neighborhood of $\theta_\star$. They encode identifiability and local curvature. Bounded features (**A2**) imply $L_0 \leq \mathbb{E}_{p_\theta}[\|\phi(Y)\|_2^2] \leq R^2$. A positive $m_0$ requires that $\phi(Y)$ under $p_{\theta_\star}$ has non-degenerate covariance.

*Remark* 3 (Preference data (BT/PL) as a special case)**.** If the learner does not observe i.i.d. draws from $p_{\theta_\star}$ but instead observes choices or pairwise preferences generated by a strongly identifiable random-utility model (e.g., Plackett–Luce with logits $\theta^\top \phi(y)$), an analogous argument applies. The MLE $\hat{\theta}_n$ still satisfies a bound of the form $\|\hat{\theta}_n - \theta_\star\|_2 \leq C\sqrt{(k + \log(1/\delta))/n}$ under standard connectivity and bounded-range assumptions for the comparison design, with $m_0$ replaced by the minimal Fisher information eigenvalue induced by that design. Lemma 3 then yields the same $\tilde{O}(\sqrt{k/n})$ TV rate for the induced per-query response distribution.

*Remark* 4 (Beyond bounded features)**.** If $\|\phi(Y)\|_2$ is unbounded but sub-Gaussian under $p_{\theta_\star}$, the concentration in Lemma 2 holds with $R$ replaced by an appropriate sub-Gaussian proxy, and the proof proceeds unchanged. If only second moments are finite, one can use Bernstein-type bounds with slightly modified logarithmic factors.

# F   PROOF OF THEOREM 2

Throughout, $\Phi$ and $\phi$ denote the standard normal CDF and PDF. We first prove a useful Gaussian tail bound via a short integration-by-parts argument (included here for completeness and rigor), and then establish parts (i)–(ii).

## F.1   EXACT RELIABILITY UNDER SPACING–SCALED GAUSSIAN NOISE

Let $k$ be even and $n = k/2$. Assume the *deterministic uniform spacing* model:

$$s_{(i+1)} - s_{(i)} \equiv \Delta := \frac{1}{k}, \qquad i = 1, \dots, k-1. \tag{18}$$

The reward model produces noisy scores

$$\hat{s}_i \;=\; s_i + \varepsilon_i, \qquad \varepsilon_1, \dots, \varepsilon_k \overset{\text{i.i.d.}}{\sim} \mathcal{N}(0, \sigma^2), \qquad \sigma \;\le\; \frac{\beta}{k}, \tag{19}$$

for a fixed constant $\beta > 0$ independent of $k$. Define the *exact* MPO correctness event

$$\text{MPO\_corr} \iff \max_{1 \le i \le n} \hat{s}_{(i)} < \min_{n+1 \le j \le k} \hat{s}_{(j)}. \tag{20}$$

**Theorem 4** (Exact MPO lower bound uniform in $k$; PL exponential fragility)**.** Under the assumptions above,

**(i) MPO (failure probability).**   the MPO failure probability admits a $k$–uniform bound

$$\Pr\big(\text{MPO\_corr}^c\big) \;\le\; C(\beta) := \sum_{m=1}^{\infty} m\, \Phi\!\left(-\frac{m}{\sqrt{2}\,\beta}\right) \;<\; \infty, \tag{21}$$

hence

$$\Pr\big(\text{MPO\_corr}\big) \;\ge\; 1 - C(\beta), \tag{22}$$

where $C(\beta)$ depends only on $\beta$ (and not on $k$). In particular, for any fixed $\beta > 0$, $\inf_k \Pr(\text{MPO\_corr}) \ge 1 - C(\beta) > 0$.

**(ii) PL (full strict order).**   Let

$$q_\beta \;:=\; \Phi\!\left(\frac{1}{\sqrt{2}\,\beta}\right) \in \left(\tfrac{1}{2}, 1\right).$$

Then the PL correctness probability satisfies the disjoint-pairs thinning bound

$$\Pr\big(\hat{s}_{(1)} < \cdots < \hat{s}_{(k)}\big) \;\le\; \big(q_\beta\big)^{\lfloor (k-1)/2 \rfloor} \;=\; \exp\big(-\gamma(\beta)\, k + O(1)\big)$$
$$\gamma(\beta) := \tfrac{1}{2}\big(-\log q_\beta\big) > 0.$$

Thus, under spacing–scaled Gaussian noise $\sigma \le \beta/k$, the exact MPO correctness probability is bounded away from $0$ uniformly in $k$, while PL correctness decays exponentially in $k$.

**Lemma 4** (Gaussian upper tail with $1/t$ prefactor)**.** If $Z \sim \mathcal{N}(0,1)$ and $t > 0$, then

$$\Pr(Z \ge t) \;\le\; \frac{\phi(t)}{t} \;=\; \frac{1}{t\sqrt{2\pi}}\, e^{-t^2/2}. \tag{23}$$

*Proof.* Using $\phi'(x) = -x\phi(x)$ and integrating by parts,

$$\Pr(Z \ge t) = \int_t^\infty \phi(x)\, dx = \left[-\frac{\phi(x)}{x}\right]_t^\infty - \int_t^\infty \frac{\phi(x)}{x^2}\, dx \le \frac{\phi(t)}{t},$$

since $\phi(x)/x \to 0$ as $x \to \infty$ and the final integral is nonnegative.  $\square$

### F.2 EXACT MPO LOWER BOUND (UNIFORM IN K)

Fix $i \le n$ and $j > n$ with *rank gap* $m := j - i \ge 1$. Uniform spacing gives

$$s_{(j)} - s_{(i)} = m\Delta = \frac{m}{k}.$$

With $\varepsilon$ i.i.d. $\mathcal{N}(0, \sigma^2)$ and $\sigma \le \beta/k$, the noisy gap satisfies

$$\hat{s}_{(j)} - \hat{s}_{(i)} = (s_{(j)} - s_{(i)}) + (\varepsilon_{(j)} - \varepsilon_{(i)}) \sim \mathcal{N}(m\Delta, 2\sigma^2),$$

so the *pairwise flip probability* is

$$\Pr(\hat{s}_{(j)} \le \hat{s}_{(i)}) = \Phi\left(-\frac{m\Delta}{\sqrt{2}\,\sigma}\right) \le \Phi\left(-\frac{m/k}{\sqrt{2}\,(\beta/k)}\right) = \Phi\left(-\frac{m}{\sqrt{2}\,\beta}\right). \tag{24}$$

For a fixed $m$, there are exactly $m$ cross–half pairs with gap $m$:

$$(i, j) \in \{(n - m + 1, n + 1), \dots, (n, n + m)\}.$$

Therefore, by a union bound over all cross–half pairs,

$$\Pr(\text{MPO\_corr}^c) \le \sum_{m=1}^{n} \sum_{\substack{i \le n,\, j > n \\ j - i = m}} \Pr(\hat{s}_{(j)} \le \hat{s}_{(i)})$$

$$\le \sum_{m=1}^{n} m\,\Phi\left(-\frac{m}{\sqrt{2}\,\beta}\right)$$

$$\le \sum_{m=1}^{\infty} m\,\Phi\left(-\frac{m}{\sqrt{2}\,\beta}\right)$$

$$=: C(\beta).$$

By Lemma 4 with $t = m/(\sqrt{2}\,\beta)$,

$$m\,\Phi\left(-\frac{m}{\sqrt{2}\,\beta}\right) \le m \cdot \frac{\phi\left(\frac{m}{\sqrt{2}\,\beta}\right)}{\frac{m}{\sqrt{2}\,\beta}} = \frac{\beta}{\sqrt{\pi}}\,\exp\left(-\frac{m^2}{4\beta^2}\right).$$

Hence $C(\beta)$ is finite for every fixed $\beta > 0$ and depends only on $\beta$:

$$C(\beta) \le \frac{\beta}{\sqrt{\pi}} \sum_{m=1}^{\infty} e^{-m^2/(4\beta^2)} \le \frac{\beta}{\sqrt{\pi}}\, e^{-1/(4\beta^2)} + \frac{\beta}{\sqrt{\pi}} \int_{1}^{\infty} e^{-x^2/(4\beta^2)}\, dx$$

$$= \frac{\beta}{\sqrt{\pi}}\, e^{-1/(4\beta^2)} + \beta\,\text{erfc}\left(\frac{1}{2\beta}\right) =: U(\beta) < \infty. \tag{25}$$

Since $U(\beta)$ is monotone increasing in $\beta$ and $U(1) < 1$ (numerically, $U(1) \approx 0.919$), we have $C(\beta) \le U(\beta) < 1$ for all $\beta \le 1$. In particular, for any fixed $\beta \le 1$,

$$\inf_k \Pr(\text{MPO\_corr}) \ge 1 - U(\beta) > 0.$$

This proves part (i).

### F.3 PL EXPONENTIAL FRAGILITY (DISJOINT-PAIRS THINNING)

Consider the disjoint adjacent pairs $(1, 2), (3, 4), \dots$, totaling $\lfloor (k - 1)/2 \rfloor$ pairs. These events are independent because they involve disjoint noise variables. Uniform spacing and $\sigma \le \beta/k$ give, for any adjacent pair,

$$\Pr(\hat{s}_{(i+1)} > \hat{s}_{(i)}) = \Pr\left(\varepsilon_{(i+1)} - \varepsilon_{(i)} > -\Delta\right) = \Phi\left(\frac{\Delta}{\sqrt{2}\,\sigma}\right) \ge \Phi\left(\frac{1}{\sqrt{2}\,\beta}\right) =: q_\beta \in (\tfrac{1}{2}, 1).$$

If the full strict order holds, then each of these disjoint events holds; thus

$$\Pr\left(\hat{s}_{(1)} < \cdots < \hat{s}_{(k)}\right) \le (q_\beta)^{\lfloor (k-1)/2 \rfloor} = \exp\left(-\gamma(\beta)\, k + O(1)\right), \qquad \gamma(\beta) = \tfrac{1}{2}(-\log q_\beta) > 0,$$

proving part (ii). $\qquad\square$

(See (Vershynin, 2018), Thm. 2.1.1 for Gaussian tail behavior; we also included a self-contained derivation of equation 23.)

# G   DIFFERENTIATING THE GROUP CONTRASTIVE LOSS FROM INFONCE LOSS

In this subsection, we compare our proposed weighted contrastive loss function with the InfoNCA loss function. We present both loss functions, derive their gradients rigorously, and characterize their stationary points. Based on this characterization, we discuss the properties of the convergence points in terms of what the models learn and their alignment with human preferences.

## G.1   DEFINITIONS OF LOSS FUNCTIONS

**InfoNCA Loss Function**   The InfoNCA loss function is defined as:

$$L_{\text{InfoNCA}} = -\sum_{i=1}^{K} p_i^{\text{target}} \log p_i^{\text{model}},$$

where $p_i^{\text{target}}$ represents the target probability for the $i$-th response, calculated as

$$p_i^{\text{target}} = \frac{e^{r(x,y_i)/\alpha}}{\sum_{j=1}^{K} e^{r(x,y_j)/\alpha}},$$

and $p_i^{\text{model}}$ denotes the model's predicted probability for the $i$-th response, given by

$$p_i^{\text{model}} = \frac{e^{s_\theta(y_i|x)}}{\sum_{j=1}^{K} e^{s_\theta(y_j|x)}}.$$

In this context, $x$ is the instruction or prompt provided to the model, and $\{y_i\}_{i=1}^{K}$ represents a set of $K$ responses generated for the instruction $x$. The term $r(x, y_i)$ is the reward associated with the response $y_i$, while $s_\theta(y_i \mid x) = \log\left(P_\theta(y_i \mid x)/P_{\text{ref}}(y_i \mid x)\right)$ is the score for response $y_i$. The parameter $\alpha$ serves as a temperature parameter that controls the influence of the reward, and $K$ is the total number of responses considered for the instruction $x$.

**Weighted Contrastive Loss Function**   Our proposed weighted contrastive loss function is expressed as:

$$L_{\text{weighted}} = -\log\left(\frac{\sum_{i \in Y^+} w_i e^{s_\theta(y_i|x)}}{\sum_{j=1}^{K} w_j e^{s_\theta(y_j|x)}}\right),$$

where $Y^+$ is the set of positive responses with rewards above the mean, defined as $Y^+ = \{y_i \mid S_i > S_{\text{mean}}\}$. Each response $y_i$ is assigned a weight $w_i = e^{\alpha \delta_i}$, where $\delta_i$ is the deviation of the reward score $S_i$ from the mean reward score $S_{\text{mean}}$. Specifically, $\delta_i = S_i - S_{\text{mean}}$ for responses in $Y^+$ and $\delta_i = S_{\text{mean}} - S_i$ for responses not in $Y^+$. The mean reward score $S_{\text{mean}}$ is calculated as

$$S_{\text{mean}} = \frac{1}{K} \sum_{j=1}^{K} S_j,$$

where $K$ is the total number of responses for the query $x$. The term $s_\theta(y_i \mid x)$ denotes the model's logit for response $y_i$, and $\alpha$ is a scaling hyperparameter that controls the influence of the deviation $\delta_i$.

## G.2 GRADIENT ANALYSIS

To understand how each loss function influences the model during training, we derive the gradients with respect to the model logits $s_\theta(y_i \mid x)$ for both methods.

**Gradient of InfoNCA Loss**

**Lemma 5.** The gradient of the InfoNCA loss with respect to the model logits $s_\theta(y_i \mid x)$ is:

$$\frac{\partial L_{\text{InfoNCA}}}{\partial s_\theta(y_i \mid x)} = p_i^{\text{model}} - p_i^{\text{target}}. \tag{26}$$

*Proof.* The InfoNCA loss is:

$$L_{\text{InfoNCA}} = -\sum_{k=1}^{K} p_k^{\text{target}} \log p_k^{\text{model}}. \tag{27}$$

Our goal is to compute $\frac{\partial L_{\text{InfoNCA}}}{\partial s_\theta(y_i|x)}$.

Since $p_k^{\text{target}}$ does not depend on $s_\theta(y_i \mid x)$ (the rewards are constants with respect to the model parameters), the derivative only affects the terms involving $p_k^{\text{model}}$.

First, express $\log p_k^{\text{model}}$ explicitly:

$$\log p_k^{\text{model}} = s_\theta(y_k \mid x) - \log\left(\sum_{j=1}^{K} e^{s_\theta(y_j|x)}\right). \tag{28}$$

Now, compute the derivative of $\log p_k^{\text{model}}$ with respect to $s_\theta(y_i \mid x)$:

$$\frac{\partial \log p_k^{\text{model}}}{\partial s_\theta(y_i \mid x)} = \frac{\partial s_\theta(y_k \mid x)}{\partial s_\theta(y_i \mid x)} - \frac{\partial}{\partial s_\theta(y_i \mid x)} \log\left(\sum_{j=1}^{K} e^{s_\theta(y_j|x)}\right). \tag{29}$$

Compute each term separately.

First term:

$$\frac{\partial s_\theta(y_k \mid x)}{\partial s_\theta(y_i \mid x)} = \delta_{ik}, \tag{30}$$

where $\delta_{ik}$ is the Kronecker delta, equal to 1 if $i = k$ and 0 otherwise.

Second term:

Let $Z = \sum_{j=1}^{K} e^{s_\theta(y_j|x)}$. Then,

$$\frac{\partial}{\partial s_\theta(y_i \mid x)} \log Z = \frac{1}{Z} \frac{\partial Z}{\partial s_\theta(y_i \mid x)}. \tag{31}$$

Compute $\frac{\partial Z}{\partial s_\theta(y_i|x)}$:

$$\frac{\partial Z}{\partial s_\theta(y_i \mid x)} = e^{s_\theta(y_i|x)}. \tag{32}$$

Therefore,

$$\frac{\partial}{\partial s_\theta(y_i \mid x)} \log Z = \frac{e^{s_\theta(y_i \mid x)}}{Z} = p_i^{\text{model}}. \tag{33}$$

Putting it all together:

$$\frac{\partial \log p_k^{\text{model}}}{\partial s_\theta(y_i \mid x)} = \delta_{ik} - p_i^{\text{model}}. \tag{34}$$

Now, compute the gradient of the loss:

$$\frac{\partial L_{\text{InfoNCA}}}{\partial s_\theta(y_i \mid x)} = -\sum_{k=1}^{K} p_k^{\text{target}} \frac{\partial \log p_k^{\text{model}}}{\partial s_\theta(y_i \mid x)} \tag{35}$$

$$= -\sum_{k=1}^{K} p_k^{\text{target}} \left( \delta_{ik} - p_i^{\text{model}} \right) \tag{36}$$

$$= -\left( p_i^{\text{target}} - p_i^{\text{model}} \sum_{k=1}^{K} p_k^{\text{target}} \right). \tag{37}$$

Since $\sum_{k=1}^{K} p_k^{\text{target}} = 1$, we have:

$$\sum_{k=1}^{K} p_k^{\text{target}} = 1 \implies \sum_{k=1}^{K} p_k^{\text{target}} = 1. \tag{38}$$

Therefore,

$$\frac{\partial L_{\text{InfoNCA}}}{\partial s_\theta(y_i \mid x)} = -\left( p_i^{\text{target}} - p_i^{\text{model}} \cdot 1 \right) = p_i^{\text{model}} - p_i^{\text{target}}. \tag{39}$$

$\square$

**Gradient of Weighted Contrastive Loss**

**Lemma 6.** The gradient of the weighted contrastive loss with respect to the model logits $s_\theta(y_i \mid x)$ is:

$$\frac{\partial L_{\text{weighted}}}{\partial s_\theta(y_i \mid x)} = p_i^{\text{weighted}} - p_i^{\text{pos}} \tag{40}$$

where:

$$p_i^{\text{weighted}} = \frac{w_i e^{s_\theta(y_i \mid x)}}{\sum_{j=1}^{K} w_j e^{s_\theta(y_j \mid x)}}, \qquad p_i^{\text{pos}} = \frac{w_i e^{s_\theta(y_i \mid x)}}{\sum_{k \in Y^+} w_k e^{s_\theta(y_k \mid x)}} \cdot \mathbb{I}_{y_i \in Y^+}, \tag{41}$$

and $\mathbb{I}_{y_i \in Y^+}$ is the indicator function, equal to 1 if $y_i \in Y^+$ and 0 otherwise.

*Proof.* Let us denote:

$$A = \sum_{k \in Y^+} w_k e^{s_\theta(y_k \mid x)}, \quad Z = \sum_{j=1}^{K} w_j e^{s_\theta(y_j \mid x)}. \tag{42}$$

The weighted contrastive loss is:

$$L_{\text{weighted}} = -\log\left(\frac{A}{Z}\right) = -\log A + \log Z. \tag{43}$$

Compute the derivative with respect to $s_\theta(y_i \mid x)$:

$$\frac{\partial L_{\text{weighted}}}{\partial s_\theta(y_i \mid x)} = -\frac{1}{A}\frac{\partial A}{\partial s_\theta(y_i \mid x)} + \frac{1}{Z}\frac{\partial Z}{\partial s_\theta(y_i \mid x)}. \tag{44}$$

Compute $\frac{\partial A}{\partial s_\theta(y_i \mid x)}$:

$$\frac{\partial A}{\partial s_\theta(y_i \mid x)} = w_i e^{s_\theta(y_i \mid x)} \cdot \mathbb{I}_{y_i \in Y^+}. \tag{45}$$

Compute $\frac{\partial Z}{\partial s_\theta(y_i \mid x)}$:

$$\frac{\partial Z}{\partial s_\theta(y_i \mid x)} = w_i e^{s_\theta(y_i \mid x)}. \tag{46}$$

Substitute back into the gradient:

$$\frac{\partial L_{\text{weighted}}}{\partial s_\theta(y_i \mid x)} = -\frac{1}{A} w_i e^{s_\theta(y_i \mid x)} \cdot \mathbb{I}_{y_i \in Y^+} + \frac{1}{Z} w_i e^{s_\theta(y_i \mid x)} \tag{47}$$

$$= w_i e^{s_\theta(y_i \mid x)} \left(\frac{1}{Z} - \frac{\mathbb{I}_{y_i \in Y^+}}{A}\right). \tag{48}$$

Recognize that:

$$p_i^{\text{weighted}} = \frac{w_i e^{s_\theta(y_i \mid x)}}{Z}, \quad p_i^{\text{pos}} = \frac{w_i e^{s_\theta(y_i \mid x)}}{A} \cdot \mathbb{I}_{y_i \in Y^+}. \tag{49}$$

Therefore:

$$\frac{\partial L_{\text{weighted}}}{\partial s_\theta(y_i \mid x)} = p_i^{\text{weighted}} - p_i^{\text{pos}}. \tag{50}$$

Since $p_i^{\text{pos}} = 0$ when $y_i \notin Y^+$, we have:

$$\frac{\partial L_{\text{weighted}}}{\partial s_\theta(y_i \mid x)} = \begin{cases} p_i^{\text{weighted}} - p_i^{\text{pos}}, & \text{if } y_i \in Y^+, \\ p_i^{\text{weighted}} - 0 = p_i^{\text{weighted}}, & \text{if } y_i \in Y^-. \end{cases} \tag{51}$$

However, this suggests that the gradient is always positive for negative examples. In other words, given $w_i$ and $Z$ are positive, $e^{s_\theta(y_i \mid x)}$ keeps increasing. But note that $s_\theta(y_i \mid x) = -\log(P_\theta(y_i \mid x))$. Hence $\frac{1}{P_\theta(y_i \mid x)}$ keeps increasing implying that $P_\theta(y_i \mid x)$ keeps decreasing. i.e. at the stationary point, $P_\theta(y_i \mid x) \to 0$ for all negative examples, $y_i \in Y^-$.

Now let us examine the positive examples. The gradient simplifies to $w_i e^{s_\theta(y_i \mid x)} \left(\frac{1}{Z} - \frac{1}{A}\right)$. Since $Z \geq A$, $\frac{1}{Z} \leq \frac{1}{A}$. Hence the gradient term with respect to $s_\theta(y_i \mid x)$ is negative. Notice that $e^{s_\theta(y_i \mid x)} = \frac{1}{P_\theta(y_i \mid x)}$. A negative gradient implies that $\frac{1}{P_\theta(y_i \mid x)}$ decreases, implying that $P_\theta(y_i \mid x)$ increases for all positive examples $y_i \in Y^+$.

$\square$

We now provide the gradients directly in terms of $P_\theta(y_j \mid x)$ instead of the scores $s_\theta(y_j \mid x)$, for easy interpretibility in terms of the probabilities.

**Lemma 7.** Let the weighted contrastive loss be defined as:

$$L_{\text{weighted}} = -\log\left(\frac{V}{U}\right) = -\log V + \log U,$$

where

$$U = \sum_{j=1}^{K} u_j P_\theta(y_j \mid x), \quad V = \sum_{i \in Y^+} u_i P_\theta(y_i \mid x),$$

and

$$u_i = \frac{w_i}{P_{\text{ref}}(y_i \mid x)},$$

with $w_i = e^{\alpha\delta_i}$, $P_\theta(y_i \mid x)$ being the model probability for response $y_i$, and $P_{\text{ref}}(y_i \mid x)$ being the reference model probability.

Then, the gradient of the weighted contrastive loss with respect to $P_\theta(y_i \mid x)$ is given by:

- For positive examples ($y_i \in Y^+$):

$$\frac{\partial L_{\text{weighted}}}{\partial P_\theta(y_i \mid x)} = u_i\left(\frac{1}{U} - \frac{1}{V}\right), \tag{52}$$

- For negative examples ($y_i \notin Y^+$):

$$\frac{\partial L_{\text{weighted}}}{\partial P_\theta(y_i \mid x)} = \frac{u_i}{U}. \tag{53}$$

*Proof.* Using the score function $s_\theta(y_i \mid x) = \log\left(\frac{P_\theta(y_i \mid x)}{P_{\text{ref}}(y_i \mid x)}\right)$, we have $e^{s_\theta(y_i \mid x)} = \frac{P_\theta(y_i \mid x)}{P_{\text{ref}}(y_i \mid x)}$.

The weighted contrastive loss becomes:

$$L_{\text{weighted}} = -\log\left(\frac{\sum_{i \in Y^+} w_i e^{s_\theta(y_i \mid x)}}{\sum_{j=1}^{K} w_j e^{s_\theta(y_j \mid x)}}\right) = -\log\left(\frac{\sum_{i \in Y^+} w_i \frac{P_\theta(y_i \mid x)}{P_{\text{ref}}(y_i \mid x)}}{\sum_{j=1}^{K} w_j \frac{P_\theta(y_j \mid x)}{P_{\text{ref}}(y_j \mid x)}}\right) = -\log\left(\frac{V}{U}\right),$$

where $u_i = \frac{w_i}{P_{\text{ref}}(y_i \mid x)}$, $V = \sum_{i \in Y^+} u_i P_\theta(y_i \mid x)$, and $U = \sum_{j=1}^{K} u_j P_\theta(y_j \mid x)$.

We compute the gradient of $L_{\text{weighted}}$ with respect to $P_\theta(y_i \mid x)$:

$$\frac{\partial L_{\text{weighted}}}{\partial P_\theta(y_i \mid x)} = -\frac{1}{V} \cdot \frac{\partial V}{\partial P_\theta(y_i \mid x)} + \frac{1}{U} \cdot \frac{\partial U}{\partial P_\theta(y_i \mid x)}.$$

**Case 1:** For $y_i \in Y^+$:

$$\frac{\partial V}{\partial P_\theta(y_i \mid x)} = u_i, \quad \frac{\partial U}{\partial P_\theta(y_i \mid x)} = u_i.$$

Thus,

$$\frac{\partial L_{\text{weighted}}}{\partial P_\theta(y_i \mid x)} = -\frac{u_i}{V} + \frac{u_i}{U} = u_i\left(\frac{1}{U} - \frac{1}{V}\right).$$

**Case 2:** For $y_i \notin Y^+$:

$$\frac{\partial V}{\partial P_\theta(y_i \mid x)} = 0, \quad \frac{\partial U}{\partial P_\theta(y_i \mid x)} = u_i.$$

Thus,

$$\frac{\partial L_{\text{weighted}}}{\partial P_\theta(y_i \mid x)} = 0 + \frac{u_i}{U} = \frac{u_i}{U}.$$

$\square$

**Corollary 1.** The sign of the gradient indicates the optimization direction:

- For positive examples ($y_i \in Y^+$), since $V \leq U$, we have $\dfrac{1}{U} - \dfrac{1}{V} \leq 0$. Therefore, the gradient $\dfrac{\partial L_{\text{weighted}}}{\partial P_\theta(y_i \mid x)} \leq 0$, and minimizing $L_{\text{weighted}}$ involves **increasing** $P_\theta(y_i \mid x)$.

- For negative examples ($y_i \notin Y^+$), the gradient $\dfrac{\partial L_{\text{weighted}}}{\partial P_\theta(y_i \mid x)} > 0$, and minimizing $L_{\text{weighted}}$ involves **decreasing** $P_\theta(y_i \mid x)$.

*Proof.* As established in the lemma:

**For positive examples** ($y_i \in Y^+$): Since $V = \sum_{i \in Y^+} u_i P_\theta(y_i \mid x)$ and $U = V + \sum_{j \notin Y^+} u_j P_\theta(y_j \mid x)$, it follows that $V \leq U$ and thus $\dfrac{1}{U} - \dfrac{1}{V} \leq 0$.

Therefore, the gradient:

$$\frac{\partial L_{\text{weighted}}}{\partial P_\theta(y_i \mid x)} = u_i \left( \frac{1}{U} - \frac{1}{V} \right) \leq 0.$$

A negative gradient indicates that increasing $P_\theta(y_i \mid x)$ will decrease $L_{\text{weighted}}$. Hence, to minimize the loss, we should **increase** $P_\theta(y_i \mid x)$ for positive examples.

**For negative examples** ($y_i \notin Y^+$): The gradient is:

$$\frac{\partial L_{\text{weighted}}}{\partial P_\theta(y_i \mid x)} = \frac{u_i}{U} > 0,$$

since $u_i > 0$ and $U > 0$. A positive gradient indicates that decreasing $P_\theta(y_i \mid x)$ will decrease $L_{\text{weighted}}$. Therefore, to minimize the loss, we should **decrease** $P_\theta(y_i \mid x)$ for negative examples. $\qquad\square$

## H    CHARACTERIZATION OF STATIONARY POINTS

We now characterize the stationary points of both loss functions.

### H.1    STATIONARY POINTS OF THE INFONCA LOSS FUNCTION

**Theorem 5.** For the InfoNCA loss, the stationary points occur when:

$$p_i^{\text{model}} = p_i^{\text{target}}, \quad \forall i \in \{1, \ldots, K\}. \tag{54}$$

*Proof.* Stationary points are defined by the condition:

$$\frac{\partial L_{\text{InfoNCA}}}{\partial s_\theta(y_i \mid x)} = 0, \quad \forall i. \tag{55}$$

From the gradient:

$$\frac{\partial L_{\text{InfoNCA}}}{\partial s_\theta(y_i \mid x)} = p_i^{\text{model}} - p_i^{\text{target}}, \tag{56}$$

setting the gradient to zero yields:

$$p_i^{\text{model}} = p_i^{\text{target}}, \quad \forall i. \tag{57}$$

$$\qquad\square$$

*Remark* 5. This stationary point is suboptimal because $p_i^{\text{model}}$ expands to:

$$p_i^{\text{model}} = \frac{e^{\log P_\theta(y_i|x) - \log P_{\text{ref}}(y_i|x)}}{\sum_{j=1}^{K} e^{\log P_\theta(y_j|x) - \log P_{\text{ref}}(y_j|x)}}$$

Rather than equating the soft-max of the difference between $\log P_\theta(y_i|x)$ and $\log P_{\text{ref}}(y_i|x)$ to $p_i^{\text{target}}$, optimality may require directly setting $\log P_\theta(y|x)$ to match the softmax of the target scores.

### H.2 STATIONARY POINTS OF THE WEIGHTED CONTRASTIVE LOSS UNDER SIMPLIFYING ASSUMPTIONS

**Lemma 8.** Consider the weighted contrastive loss function in a simplified scenario with the following conditions: There are $N^+$ positive examples, each with weight $w^+$, and $N^-$ negative examples, each with weight $w^-$. All positive examples have the same score $s^{(t)}$ at iteration $t$, and all negative examples have the same score $s^{(t)}$ at iteration $t$. Then, the update rule for the score $s^{(t)}$ of the positive examples at iteration $t+1$ is given by

$$s^{(t+1)} = s^{(t)} + \eta \left( \frac{N^- w^-}{N^+ (N^+ w^+ + N^- w^-)} \right), \tag{58}$$

where $\eta$ is the learning rate.

*Proof.* Let $Y^+$ denote the set of positive examples and $Y^-$ the set of negative examples, with $N^+$ and $N^-$ examples respectively for a total of $K = N^+ + N^-$ examples. With weights $w^+$ and $w^-$ assigned to positive and negative examples respectively, and logits $s^{(t)}$ for both classes at timestep $t$, the weighted contrastive loss function is defined as:

$$L_{\text{weighted}}(\theta) = -\log \left( \frac{\sum_{i \in Y^+} w_i e^{s_i}}{\sum_{j=1}^{K} w_j e^{s_j}} \right), \tag{59}$$

where $w_i = w^+$ and $s_i = s^{(t)}$ for $i \in Y^+$, and $w_j = w^-$ and $s_j = s^{(t)}$ for $j \in Y^-$.

Compute the numerator $A$ and the denominator $Z$ of the loss function:

$$A = \sum_{i \in Y^+} w_i e^{s_i} = N^+ w^+ e^{s^{(t)}}, \tag{60}$$

$$Z = \sum_{j=1}^{K} w_j e^{s_j} = N^+ w^+ e^{s^{(t)}} + N^- w^- e^{s^{(t)}} = e^{s^{(t)}} (N^+ w^+ + N^- w^-). \tag{61}$$

For positive examples $i \in Y^+$, the weighted probability $p_i^{\text{weighted}}$ and the positive probability $p_i^{\text{pos}}$ are:

$$p_i^{\text{weighted}} = \frac{w^+ e^{s^{(t)}}}{Z} = \frac{w^+}{N^+ w^+ + N^- w^-}, \tag{62}$$

$$p_i^{\text{pos}} = \frac{w^+ e^{s^{(t)}}}{A} = \frac{w^+}{N^+ w^+} = \frac{1}{N^+}. \tag{63}$$

The gradient of the loss with respect to $s^{(t)}$ for positive examples is:

$$\frac{\partial L_{\text{weighted}}}{\partial s^{(t)}} = p_i^{\text{weighted}} - p_i^{\text{pos}} = \frac{w^+}{N^+ w^+ + N^- w^-} - \frac{1}{N^+}. \tag{64}$$

To simplify this expression, we find a common denominator $D = N^+(N^+w^+ + N^-w^-)$:

$$\frac{\partial L_{\text{weighted}}}{\partial s^{(t)}} = \frac{w^+N^+ - (N^+w^+ + N^-w^-)}{D} \tag{65}$$

$$= \frac{w^+N^+ - N^+w^+ - N^-w^-}{N^+(N^+w^+ + N^-w^-)} \tag{66}$$

$$= \frac{-N^-w^-}{N^+(N^+w^+ + N^-w^-)}. \tag{67}$$

The update rule for $s^{(t)}$ is then:

$$s^{(t+1)} = s^{(t)} - \eta \frac{\partial L_{\text{weighted}}}{\partial s^{(t)}} = s^{(t)} + \eta \left( \frac{N^-w^-}{N^+(N^+w^+ + N^-w^-)} \right). \tag{68}$$

This completes the proof. $\square$

**Corollary 2.** Assuming the initial scores are zero ($s^{(0)} = 0$), the score $s^{(t)}$ of the positive examples at iteration $t$ is given by

$$s^{(t)} = t\eta \left( \frac{N^-w^-}{N^+(N^+w^+ + N^-w^-)} \right). \tag{69}$$

*Proof.* From the update rule established in the lemma,

$$s^{(t+1)} = s^{(t)} + c, \tag{70}$$

where

$$c = \eta \left( \frac{N^-w^-}{N^+(N^+w^+ + N^-w^-)} \right). \tag{71}$$

Since $s^{(0)} = 0$, we have

$$s^{(1)} = s^{(0)} + c = c, \tag{72}$$

$$s^{(2)} = s^{(1)} + c = 2c, \tag{73}$$

$$\vdots \tag{74}$$

$$s^{(t)} = tc. \tag{75}$$

Substituting $c$ back into the expression, we obtain

$$s^{(t)} = t\eta \left( \frac{N^-w^-}{N^+(N^+w^+ + N^-w^-)} \right). \tag{76}$$

$\square$

**Corollary 3.** In the special case where there is one positive example ($N^+ = 1$) and one negative example ($N^- = 1$), and the weights are $w^+ = w^- = 1$ (as in Direct Preference Optimization), the score $s^{(t)}$ at iteration $t$ is:

$$s^{(t)} = \frac{\eta t}{2}. \tag{77}$$

*Proof.* Substituting $N^+ = N^- = 1$ and $w^+ = w^- = 1$ into the expression for $s^{(t)}$:

$$s^{(t)} = t\eta \left( \frac{1 \times 1}{1 \times (1 \times 1 + 1 \times 1)} \right) \tag{78}$$

$$= t\eta \left( \frac{1}{1 \times (1 + 1)} \right) \tag{79}$$

$$= t\eta \left( \frac{1}{2} \right) \tag{80}$$

$$= \frac{\eta t}{2}. \tag{81}$$

$\square$

**Lemma 9.** Consider the general case where positive examples may have different weights $w_i^+$, and each positive example $i$ has its own score $s_i^{(t)}$ at iteration $t$. Assuming initial scores $s_i^{(0)} = 0$ for all positive examples, the score $s_i^{(t)}$ of positive example $i$ at iteration $t$, up to a linear approximation, is given by

$$s_i^{(t)} = t\eta w_i^+ \left(\frac{B_0}{A_0 Z_0}\right), \tag{82}$$

where $A_0 = \sum_{k \in Y^+} w_k^+$, $B_0 = \sum_{j \in Y^-} w_j^-$, $Z_0 = A_0 + B_0$, and $\eta$ is the learning rate.

*Proof.* At iteration $t = 0$, the initial scores are $s_i^{(0)} = 0$ for all $i \in Y^+$. The sums are:

$$A_0 = \sum_{k \in Y^+} w_k^+ e^{s_k^{(0)}} = \sum_{k \in Y^+} w_k^+ = W^+, \tag{83}$$

$$B_0 = \sum_{j \in Y^-} w_j^- e^{s_j^{(0)}} = \sum_{j \in Y^-} w_j^- = W^-. \tag{84}$$

The total sum is $Z_0 = A_0 + B_0 = W^+ + W^-$.

The gradient for each positive example $i$ at $t = 0$ is:

$$\frac{\partial L_{\text{weighted}}}{\partial s_i^{(0)}} = -w_i^+ e^{s_i^{(0)}} \left(\frac{B_0}{A_0 Z_0}\right) = -w_i^+ \left(\frac{B_0}{A_0 Z_0}\right). \tag{85}$$

The update rule is:

$$s_i^{(1)} = s_i^{(0)} - \eta \frac{\partial L_{\text{weighted}}}{\partial s_i^{(0)}} = \eta w_i^+ \left(\frac{B_0}{A_0 Z_0}\right). \tag{86}$$

Assuming that the term $\frac{B_0}{A_0 Z_0}$ remains approximately constant over iterations (which holds when $\eta$ is small and changes in $s_i^{(t)}$ are small), the score at iteration $t$ is:

$$s_i^{(t)} = t\eta w_i^+ \left(\frac{B_0}{A_0 Z_0}\right). \tag{87}$$

$\square$

*Remark* 6. The approximation assumes that $A_t$, $B_t$, and $Z_t$ remain close to their initial values $A_0$, $B_0$, and $Z_0$ over the iterations considered, and the score values remain small. This is reasonable for small learning rates $\eta$ and a limited number of iterations $t$.

### H.3 STATIONARY POINTS OF THE WEIGHTED CONTRASTIVE LOSS

We now analyze the stationary points of our weighted contrastive loss function.

**Lemma 10.** For the weighted contrastive loss function, the stationary point occurs when the probabilities of the negative samples approach zero, i.e.,

$$P_\theta(y_i \mid x) \to 0 \quad \text{for all } y_i \in Y^-. \tag{88}$$

*Proof.* From Lemma 6, the gradient of the weighted contrastive loss with respect to the model logits $s_\theta(y_i \mid x)$ is:

$$\frac{\partial L_{\text{weighted}}}{\partial s_\theta(y_i \mid x)} = \begin{cases} p_i^{\text{weighted}} - p_i^{\text{pos}}, & \text{if } y_i \in Y^+, \\ p_i^{\text{weighted}}, & \text{if } y_i \in Y^-. \end{cases} \tag{89}$$

At a stationary point, the gradient must be zero for all $y_i$. Consider the negative samples $y_i \in Y^-$. Setting the gradient to zero yields:

$$\frac{\partial L_{\text{weighted}}}{\partial s_\theta(y_i \mid x)} = p_i^{\text{weighted}} = 0. \tag{90}$$

Since $p_i^{\text{weighted}}$ is the normalized weighted probability of $y_i$, given by:

$$p_i^{\text{weighted}} = \frac{w_i e^{s_\theta(y_i|x)}}{\sum_{j=1}^{K} w_j e^{s_\theta(y_j|x)}},\tag{91}$$

and $w_i > 0$, the only way for $p_i^{\text{weighted}}$ to be zero is if $e^{s_\theta(y_i|x)} = 0$, which implies:

$$s_\theta(y_i \mid x) \to -\infty \quad \Longrightarrow \quad P_\theta(y_i \mid x) \to 0 \quad \text{for } y_i \in Y^-.\tag{92}$$

Similarly, for positive samples $y_i \in Y^+$, the gradient is:

$$\frac{\partial L_{\text{weighted}}}{\partial s_\theta(y_i \mid x)} = p_i^{\text{weighted}} - p_i^{\text{pos}} = 0.\tag{93}$$

This implies:

$$p_i^{\text{weighted}} = p_i^{\text{pos}}.\tag{94}$$

Since the probabilities of the negative samples approach zero, the denominator in $p_i^{\text{weighted}}$ becomes:

$$\sum_{j=1}^{K} w_j e^{s_\theta(y_j|x)} \approx \sum_{k \in Y^+} w_k e^{s_\theta(y_k|x)}.\tag{95}$$

Therefore, $p_i^{\text{weighted}} \approx p_i^{\text{pos}}$, satisfying the condition for the gradient to be zero for positive samples.

Thus, at the stationary point, the probabilities of the negative samples approach zero. $\square$

*Remark* 7. When the probabilities of the negative samples approach zero, the scores $s_\theta(y_i \mid x)$ for $y_i \in Y^-$ tend to $-\infty$. Since:

$$e^{s_\theta(y_i|x)} = \frac{P_\theta(y_i \mid x)}{P_{\text{ref}}(y_i \mid x)} \to 0,\tag{96}$$

the weighted contributions of the negative samples to the numerator and denominator of $L_{\text{weighted}}$ become negligible.

Consequently, the numerator and denominator of $L_{\text{weighted}}$ become equal:

$$\sum_{i \in Y^+} w_i e^{s_\theta(y_i|x)} \approx \sum_{j=1}^{K} w_j e^{s_\theta(y_j|x)}.\tag{97}$$

Therefore:

$$L_{\text{weighted}} = -\log\left(\frac{\sum_{i \in Y^+} w_i e^{s_\theta(y_i|x)}}{\sum_{j=1}^{K} w_j e^{s_\theta(y_j|x)}}\right) \approx -\log 1 = 0.\tag{98}$$

This implies that the loss vanishes when the probabilities of the negative samples approach zero, indicating that the model has successfully minimized the loss by focusing entirely on the positive responses.

## I    BASELINES USED FOR COMPARISON

When dealing with reward datasets where each instruction has more than two $n > 2$ responses, one common approach is to convert the data into pairwise preferences and then apply preference optimization techniques such as Direct Preference Optimization (DPO). Several strategies can be adopted for this purpose, each offering distinct trade-offs in terms of dataset richness and computational overhead.One straightforward method, as implemented by Zephyr Tunstall et al. (2023), involves selecting the response with the highest reward and pairing it with a randomly chosen response from the remaining responses for each instruction. Another variant involves pairing the

| Method | Llama3-8B-Instruct | | | Mistral-7B-Instruct | | |
|--------|-----------|--------|-----------|-----------|--------|-----------|
| | LC-WR (%) | WR (%) | Std. Err. | LC-WR (%) | WR (%) | Std. Err. |
| DPO | 41.1 | 37.8 | 1.38 | 26.8 | 24.6 | 1.39 |
| MPO | 49.0 | 50.6 | 1.52 | 28.2 | 29.4 | 1.34 |
| W-MPO | **52.1** | **52.5** | 1.46 | **29.5** | **30.6** | 1.45 |

Table 9: Comparison of MPO variants with DPO on **Llama3-8B-Instruct** and **Mistral-7B-Instruct**. We report Length-Controlled Win Rate (LC-WR), Overall Win Rate (WR), and standard error across evaluations.

highest-rewarded response with the lowest-rewarded response for each instruction, ensuring a clear distinction between preferences.Additionally, alternative baselines can be explored to enhance performance by incorporating more suboptimal responses during training. By applying DPO to combinations of responses, we can significantly expand the preference dataset and potentially achieve improved optimization. Baselines in this context are:

**DPOx$\binom{N}{2}$**: In this approach, all possible pairwise combinations of $\binom{N}{2}$ are generated, and DPO is applied to the entire combinatorial dataset. This method ensures the model is exposed to a comprehensive range of preference relationships, including those involving suboptimal responses.

**Group Relative Policy Optimization (GRPO)**: We implement GRPO in an offline setup, where both the response set (four responses per prompt) and their scalar rewards are fixed, as provided by ULTRAFEEDBACK. GRPO uses a relative advantage formulation that weighs each response based on its normalized reward gap.

**Plackett-Luce Ranking Model**: We implement the Plackett-Luce model, in which the ranking over responses is determined by the reward scores provided in ULTRAFEEDBACK. The objective maximizes the likelihood of the observed ranking via a sequential softmax over score-ordered responses.

**InfoNCA**: We adapt InfoNCA to the preference optimization setting by treating each response as a sample in a softmax-based contrastive loss, where rewards from ULTRAFEEDBACK define the similarity scores. All responses are used jointly during training.

**MPO-1vsk**: In this baseline for each prompt, the response with the highest reward is selected as the positive sample, and the remaining responses form the negative set in MPO's formulation.

All baselines are trained under a unified offline pipeline to ensure fair and consistent comparison across evaluation benchmarks.

## J EXPERIMENTAL DETAILS

**Model and Training Setting** For our off-policy experiments, we utilized the Ultrafeedback Dataset Cui et al. (2023), an instruction-following benchmark annotated by GPT-4. This dataset consists of approximately 64,000 instructions, each paired with four responses generated by different language models. GPT-4 assigned scalar rewards on a 0-to-10 scale for each response, which prior research has shown to correlate strongly with human annotations. This establishes GPT-4 ratings as a reliable and cost-efficient alternative to manual feedback.

In our broader framework, we first trained a base model (mistralai/Mistral-7B-v0.1) and (meta-llama/Meta-Llama-3-8B) on the UltraChat-200k dataset to obtain an SFT model. This SFT model, trained on open-source data, provides a transparent starting point. Subsequently, we refined the model by performing preference optimization on the UltraFeedback dataset. Once fine-tuned, the model was used for alignment. This two-step process ensures the model is well-prepared for tasks.

In our experiments, we observed that tuning hyperparameters is critical for optimizing the performance of all offline preference optimization algorithms, including DPO, SimPO, MPO and W-MPO. Hyperparameter selection is very crucial for these.For MPO, we found that setting the $\beta$ parameter in the range of 0.01 to 0.05 consistently yields strong performance. For W-MPO, we found that setting the $\beta$ parameter in the range of 0.01 to 0.05 and $\alpha$ to 10 consistently yields strong performance.

Learning rate for mistral-experiments was fixed to 3e-7 and whereas for llama it was 5e-7 for all the baselines.

For our on-policy and iterative experiments, we utilize a pretrained instruction-tuned model (meta-llama/MetaLlama-3-8B-Instruct) and (mistralai/Mistral-7B-Instruct-v0.2) , as the SFT model. These models have undergone extensive instruction tuning, making them more capable and robust compared to the SFT models used in the Base setup. However, their reinforcement learning with human feedback (RLHF) procedures remain undisclosed, making them less transparent.

To reduce distribution shift between the SFT models and the preference optimization process, we follow the approach in Meng et al. (2024) and generate the preference dataset using the same SFT models. This ensures that our setup is more aligned with an on-policy setting. Specifically, we utilize prompts from the UltraFeedback dataset Cui et al. (2023) and regenerate the resonses using the SFT models. For each prompt x, we produce 5 responses by sampling from the SFT model with a sampling temperature of 1.0 and top-p 1.0. We then use the reward model (Skywork/Skywork-Reward-Llama-3.1-8B-v0.2) Liu et al. (2024b) to score all the responses

Table 10: Hyperparameters $\beta$ and learning rate $\alpha$ across different training settings.

| Setting | Model | Iter | $\beta$ | $lr$ |
|---------|-------|------|---------|------|
| Offline | Mistral-Base (7B) | – | 0.01 | 4e-7 |
| | Llama3-Base (8B) | – | 0.01 | 5e-7 |
| Online | Mistral-Instruct (7B) | – | 0.01 | 1.5e-7 |
| | LLama3-Instruct (8B) | – | 0.01 | 3e-7 |
| Iterative | Mistral-Instruct (7B) | 1 | 0.01 | 2.5e-7 |
| | Llama3-Instruct (8B) | 1 | 0.01 | 5e-7 |
| | Mistral-Instruct (7B) | 2 | 0.01 | 2e-7 |
| | Llama3-Instruct (8B) | 2 | 0.01 | 4e-7 |
| | Mistral-Instruct (7B) | 3 | 0.01 | 1.5e-7 |
| | Llama3-Instruct (8B) | 3 | 0.01 | 3e-7 |

In our online and iterative training experiments, we found that systematic hyperparameter tuning is critical for optimizing performance across diverse datasets. Notably, the choice of the $\beta$ parameter had a substantial impact, with values in the range of 0.01 to 0.1 consistently yielding strong results. For all experiments, we adopted a fixed learning rate of 1.5e-7 for mistral-based models and 3.0e-7 for llama-based models.

All experiments employed the AdamW optimizer with a cosine annealing learning rate scheduler, incorporating a 0.01% warmup step and a total of 1 training epoch. We used 8×A100 (80GB) GPUs under DeepSpeed with distributed data parallel (DDP) training. The global batch size was set to 128, with a per-device batch size of 2 and a gradient accumulation step of 8 to ensure memory efficiency and training stability.

These findings emphasize the importance of well-calibrated hyperparameters and infrastructure settings to achieve robust and reproducible outcomes in post-training alignment tasks.

**Evaluation Benchmarks:** We evaluate our models using three widely recognized open-ended instruction-following benchmarks: MT-Bench Zheng et al. (2023), AlpacaEval 2, and Arena-Hard v0.1 Zheng et al. (2023). These benchmarks test the models' conversational versatility across a broad range of queries and are broadly utilized in the research community. AlpacaEval2 Dubois et al. (2024) includes 805 questions derived from five datasets, while MT-Bench spans eight categories with a total of 80 questions. Arena-Hard, a recently updated version of MT-Bench, focuses on 500 well-defined technical problem-solving queries. Scores are reported based on the evaluation protocols of each benchmark. For AlpacaEval2, both the raw win rate (WR) and the length-controlled win rate (LC) are reported, with LC specifically designed to mitigate biases related to model verbosity. For Arena-Hard, the win rate is reported relative to a baseline model. For MT-Bench, the average score is calculated using evaluations by GPT-4 as judge. For decoding details, we generate responses using both greedy decoding and multinomial sampling with temperatures of 0.2, 0.5, and 1.0. To address potential biases introduced by multinomial sampling at varying temperatures, we generate responses three times for each setting at different seed and average their performance across the datasets.

## K  REWARD LOSS COMPUTATION

In this section we provide the actual code used to compute the reward losses.

```python
import torch

def mpo_loss(pi_logps, ref_logps, rewards, beta, alpha):
    """
    Computes the Weighted MPO (W-MPO) loss.
    r'_theta = (beta * r_theta) + (alpha * Delta_W_abs)

    pi_logps: policy logprobs for N responses, shape (Batch_Size, N)
    ref_logps: reference logprobs for N responses, shape (Batch_Size, N)
    rewards: RAW reward labels for N responses, shape (Batch_Size, N)
    beta: Temperature parameter scaling the policy logits (r_theta)
    alpha: Weight scaling the absolute reward deviation (Delta_W_abs)
    """
    # 1. Calculate base logits (r_theta)
    base_logits = pi_logps - ref_logps

    # 2. Calculate mean_rewards and weights (Delta_W_abs)
    mean_rewards = torch.mean(rewards, dim=-1, keepdim=True)

    # We assume weighting is enabled if alpha > 0.
    if alpha > 0:
        weights = torch.abs(rewards - mean_rewards) # This is Delta_W_abs
    else:
        weights = torch.zeros_like(rewards)

    # 3. Calculate masks based on RAW rewards
    pos_mask = (rewards > mean_rewards) * 1
    neg_mask = (rewards <= mean_rewards) * 1 # Use <= for logical not

    # 4. Calculate final logits r'_theta = (beta * r_theta) + (alpha *
        Delta_W_abs)
    scaled_logits = base_logits * beta
    scaled_weights = weights * alpha

    final_logits = scaled_logits + scaled_weights

    # 5. Stabilize and compute loss
    eps = 1e-10

    # Stabilize logits by subtracting the max for numerical stability
    final_logits = final_logits - final_logits.max(dim=-1, keepdim=True)
        [0]  # Stabilize logits

    softmax_val = torch.softmax(final_logits + eps, dim=-1)

    # Calculate sum of probabilities for positive and negative sets
    pos_sum = torch.clamp(torch.sum(softmax_val * pos_mask, dim=-1), min=
        eps)
    neg_sum = torch.clamp(torch.sum(softmax_val * neg_mask, dim=-1), min=
        eps)

    # Denominator is the sum over all (positive and negative) responses
    denominator = pos_sum + neg_sum + eps * 2

    losses = -1 * torch.log(pos_sum / denominator)

    return losses.mean()
```

## L    EXAMPLES

In this section, we provide examples of user queries and responses from our model and the base DPO model.

---

**DPO vs MPO comparison**

**User Query**

*"Provide a pleasant compliment based on the quality you have been given. Truthfulness"*

**DPO Response**

You possess a rare and valuable quality of truthfulness, which shines through in everything you say and do. Your authenticity is refreshing and inspiring, and it's clear that you're committed to being honest and transparent in all your interactions. Your words carry weight because they're backed by integrity, and it's a true asset to be around someone who embodies such a high level of trustworthiness. Your truthfulness is a gift to those around you, and it's something to be deeply proud of.

**MPO Response**

Here's a pleasant compliment: "You have a rare gift for authenticity, and it's truly refreshing to be around someone as honest and transparent as you. Your words carry weight because you always speak from the heart, building trust and respect with everyone you interact with. Your truthfulness is a treasure, and it's a privilege to know someone as genuine and dependable as you!"

---

## DPO vs MPO comparison

### User Query

*"Write a to-do list based on the given information. Daily tasks in the pharmacy?"*

### DPO Response

Based on typical daily tasks in a pharmacy, here's a to-do list:
**Morning Routine**
1. Restock shelves with medications and supplies 2. Check and respond to phone calls and voicemails 3. Review and update patient medication profiles and records
**Patient Care**
1. Fill and dispense prescriptions accurately and efficiently 2. Answer patient questions and provide medication counseling 3. Monitor and manage medication therapy plans for chronic conditions
**Inventory Management**
1. Conduct inventory counts and report any discrepancies 2. Order new stock and supplies as needed 3. Manage expired or recalled medications and dispose of them properly
**Administrative Tasks**
1. Process and submit claims to insurance companies 2. Manage and reconcile cash and credit transactions 3. Update patient records and billing information
**End-of-Day Tasks**
1. Clean and sanitize work areas and equipment 2. Review and prepare for the next day's tasks 3. Document any issues or concerns that arose during the day

### MPO Response

Here's a comprehensive to-do list for daily tasks in a pharmacy:
**Morning Routine (Before Opening)**
1. **Inventory Management**: * Check inventory levels of medications, supplies, and equipment. * Restock shelves and replenish stock as needed. 2. **Preparation**: * Prepare prescription labels, bags, and packaging materials. * Set up cash register, credit card machines, and point-of-sale systems. 3. **Computer Systems**: * Log in to pharmacy management software (e.g., Meditech, Cerner) and update patient records. * Ensure all systems are functioning properly.
**During Business Hours**
1. **Fill Prescriptions**: * Process and fill patient prescriptions accurately and efficiently. * Verify prescriptions with physicians (if necessary) and communicate with patients. 2. **Customer Service**: * Assist customers with questions, concerns, or medication inquiries. * Provide guidance on medication usage, side effects, and interactions. 3. **Order Entry**: * Enter new prescriptions, refill requests, and medication orders into the system. * Manage backorders and special orders. 4. **Dispensing and Packaging**: * Package and label medications, including controlled substances. * Ensure proper storage and security of controlled substances. 5. **Administrative Tasks**: * Manage phone calls, emails, and patient inquiries. * Update patient records and maintain accurate records.
**End of Day/Before Closing**
1. **Closing Procedures**: * Count and reconcile cash, credit card transactions, and inventory. * Secure the pharmacy, including controlled substances and sensitive areas. 2. **Reporting and Documentation**: * Complete daily reports (e.g., prescription fills, inventory, and sales). * Update records and submit reports as required by regulatory authorities. 3. **Cleanliness and Organization**: * Maintain a clean and organized pharmacy environment, including workstations and storage areas.

## DPO vs MPO comparison

### User Query

*"GIVE ME AN EXECUTIVE SUMMARY OF LOTTADIGITAL.COM?"*

### DPO Response

Based on my research, LotT Digital is a digital marketing agency that specializes in providing comprehensive solutions to help businesses grow online. Here's an executive summary:

**Key Focus Areas:** 1. Digital Strategy: LotT Digital develops tailored strategies for clients, encompassing market analysis, competitor research, and goal-setting. 2. Web Development: The agency builds custom websites, e-commerce platforms, and web applications using modern technologies like WordPress, Magento, and React. 3. Digital Marketing: Services include SEO, PPC advertising (Google Ads), social media marketing, email marketing, and content creation to drive traffic and conversions. 4. Data Analytics: LotT Digital provides insights and reporting to measure campaign performance, track key metrics, and inform data-driven decision-making.

**Target Industries and Clients:** 1. Mid-to-large-sized businesses across various sectors, including e-commerce, finance, healthcare, and professional services. 2. Companies seeking to enhance their online presence, increase website traffic, and drive revenue growth.

**Unique Value Proposition:**

LotT Digital's team of experts takes a holistic approach to digital marketing, combining strategy, creativity, and technical expertise to deliver measurable results. They prioritize transparency, regular communication, and continuous optimization to ensure client success.

### MPO Response

Here's an executive summary of Lottadigital.com:

**Company Overview: Lottadigital**

Lottadigital.com is a leading digital marketing agency specializing in E-commerce Solutions, Digital Strategy, and Performance Marketing. Headquartered in the UK, the company helps online businesses thrive in a competitive landscape.

**Key Focus Areas:**

1. **E-commerce Services**: Lottadigital offers comprehensive e-commerce solutions, including website design, development, and optimization for various platforms (e.g., Shopify, Magento, BigCommerce). 2. **Digital Strategy**: They craft tailored strategies to drive business growth, leveraging expertise in SEO, PPC (Google Ads), social media marketing, and content marketing. 3. **Performance Marketing**: The agency excels in performance-based marketing, focusing on data-driven campaigns that deliver measurable ROI, such as affiliate marketing, influencer partnerships, and retargeting. 4. **Specializations**: Lottadigital has expertise in niches like fashion, beauty, health, and lifestyle, with a strong understanding of their respective markets and consumer behaviors.

**Key Benefits:**

* Data-driven approach to optimize online presence and conversions * Customized solutions for businesses seeking to improve their digital footprint * Proven track record of increasing website traffic, sales, and revenue growth * Expertise in staying up-to-date with the latest industry trends and platforms

**In summary**, Lottadigital.com is a full-service digital marketing agency that empowers e-commerce businesses to succeed by providing strategic, performance-driven solutions, tailored to their specific needs. Their expertise helps clients achieve measurable success and stay ahead in a rapidly evolving digital landscape.

# M    QUALATITIVE STUDY

## Question:

*What are the names of some famous actors that started their careers on Broadway?*

---

**DPO Response**

The Great White Way to stardom! Many renowned actors have indeed launched their careers on Broadway before transitioning to Hollywood success. Here's a list of famous actors who got their start on the Broadway stage: **1. Robert De Niro** (*The Three Sisters*, 1969) — Tony Award nominee, went on to become a legendary film icon. **2. Julia Roberts** (*Steel Magnolias*, 1987) — Broadway debut led to a successful film career. **3. Nathan Lane** (*Noises Off*, 1983) — Tony Award winner and beloved character actor. **4. Hugh Jackman** (*Oklahoma!*, 1998) — Broadway's Curly McLain led to his breakout role as Wolverine. **5. Audrey Hepburn** (*Gigi*, 1951) — Her Broadway debut earned her a Tony nomination. **6. Al Pacino** (*Does a Tiger Dream of Wings*, 1969) — Off-Broadway and Broadway credits preceded iconic film roles. **7. Idina Menzel** (*Rent*, 1996) — Her Tony-nominated performance launched her career. **8. Neil Patrick Harris** (*Assassins*, 1990) — Broadway's Stephen Weed and later Barney Stinson. **9. Christine Baranski** (*Company*, 1970) — A Tony Award winner and Broadway veteran. **10. Jake Gyllenhaal** (*This Is Our Youth*, 1994) — His Broadway debut led to a successful film career. **11. Zooey Deschanel** (*This Is Our Youth*, 1996) — Her Broadway experience paved the way for TV roles. **12. Ralph Fiennes** (*Hamlet*, 1995) — A Tony Award nominee for his Broadway performance.
These are just a few examples of the many talented actors who got their start on the Broadway stage.

---

**MPO Response**

Many talented actors have transitioned from Broadway to successful film and television careers. Here are some famous examples: **1. Julie Andrews** — Starred in Broadway productions like *My Fair Lady* and *Camelot* before becoming a Hollywood icon. **2. Hugh Jackman** — Originated the role of Curly McLain in *Oklahoma!* before starring in *X-Men*. **3. Neil Patrick Harris** — Got his start on Broadway in *Assassins* and *Rent*. **4. Idina Menzel** — Originated the role of Maureen Johnson in *Rent*. **5. Nathan Lane** — Won a Tony Award for *The Producers* before appearing in films like *The Birdcage*. **6. Bernadette Peters** — Starred in *Sunday in the Park with George* and *Into the Woods*. **7. Liev Schreiber** — Originated the role of Emcee in the Broadway revival of *Cabaret*. **8. Sutton Foster** — Got her start on Broadway in *Thoroughly Modern Millie*. **9. Zachary Quinto** — Appeared in *Angels in America* before *Star Trek*. **10. Laura Linney** — Won a Tony Award for *Sight Unseen*. **11. Alan Cumming** — Originated the role of Emcee in the revival of *Cabaret*. **12. Christine Baranski** — Won a Tony Award for *Rumors*.

---

**GPT-5 Judgment**

The **MPO response** is superior in both factual accuracy and presentation. While the DPO response adds flair through theatrical phrasing, it includes multiple factual inaccuracies—several mentioned actors did not actually start their careers on Broadway. Moreover, it is verbose and inconsistent in formatting. In contrast, the MPO response is concise, accurate, and well-organized: each listed actor has verifiable Broadway roots, and the transition from stage to screen is clearly explained.

**Gemini 3 Pro Judgement**

The MPO response is significantly better because it is far more factually accurate. The DPO response contains severe hallucinations and misinformation; for example, it incorrectly claims that Julia Roberts, Jake Gyllenhaal, and Zooey Deschanel started their careers on Broadway (Deschanel has never appeared on Broadway). Conversely, the MPO response correctly identifies actors who truly built their initial fame on the stage.

## Question:

*Can you tell me a very easy way to clean a showerhead?*

**DPO Response**

Cleaning your showerhead regularly can help remove mineral buildup. Here's a super easy way to clean your showerhead:
**Method: Vinegar Soak** 1. Remove the showerhead from the shower arm. 2. Fill a plastic bag with white vinegar (you can use a zip-top bag or a mesh produce bag). 3. Place the showerhead inside the bag, making sure it's fully submerged in the vinegar. 4. Seal the bag and let it sit for 2-3 hours or overnight. 5. After the soaking time, remove the showerhead from the bag and rinse it thoroughly with warm water. 6. Use a soft cloth or toothbrush to gently scrub away any remaining mineral deposits. 7. Reattach the showerhead to the shower arm and turn on the shower to test its water flow.
**Tips:** For tougher mineral buildup, repeat the vinegar soak. Avoid abrasive cleaners.

**MPO Response**

A simple yet essential question! Cleaning your showerhead regularly is crucial to remove mineral buildup. Here's a super easy way to clean a showerhead:
**Method: Vinegar Soak and Rinse** 1. **Disassemble (if possible):** If your showerhead has removable parts, take it apart. If not, no worries. 2. **Fill a plastic bag with vinegar:** Place the showerhead (or its parts) in a plastic bag. 3. **Add white vinegar:** Pour enough white vinegar to cover the showerhead completely. 4. **Seal and soak:** Close the bag and let it sit for 1-2 hours. 5. **Rinse thoroughly:** Remove the showerhead and rinse under running warm water. 6. **Shake and scrub (optional):** Gently shake or scrub with a toothbrush. 7. **Dry and reassemble:** Reattach parts and dry with a towel.
**Tips:** Avoid using harsh chemicals. Clean every 3-6 months depending on water quality.

**GPT-5 Judgment**

The MPO response is better because it is more engaging, structured, and user-friendly while maintaining a clear instructional flow. It begins with a short contextual sentence that emphasizes the importance of the task. The steps are detailed yet simple, adapting for different situations (e.g., detachable vs. non-detachable showerheads). In contrast, the DPO response, while correct and concise, feels more mechanical.

**Gemini 3 Pro Judgement**

The MPO response is better because it avoids the critical logical errors found in the DPO response. The DPO response suggests filling a "mesh produce bag" with liquid vinegar, which is physically impossible as the liquid would leak out immediately. MPO provides a more standard approach and avoids advising the user to perform impossible tasks.

## Question:

*How can I fix a leaking roof?*

**DPO Response**

**1. Identify the source of the leak:** Locate the area where water is entering. **2. Clear the area:** Move any furniture or debris. **3. Contain the leak:** Cover the floor with tarps. **4. Inspect and clean the roof:** Remove debris like leaves or branches. **5. Repair or replace damaged shingles:** Replace single shingles or hire a pro for extensive damage. **6. Check and repair flashing:** Inspect seals around vents and chimneys. **7. Seal around vents, skylights, and chimneys:** Recaulk if necessary. **8. Apply roofing cement or sealant:** Fill gaps and cracks. **9. Install a roof patch:** For extensive leaks. **10. Inspect and test:** Verify integrity with water. **11. Consider hiring a professional:** Recommended for safety.
**Additional Tips:** Always follow safety guidelines. Check warranty.

**MPO Response**

The age-old dilemma! Fixing a leaking roof can be daunting, but here is a comprehensive guide:
**1. Assess the leak:** Locate the source by looking for water stains or droplets. Inspect for damaged shingles or ice dams. **2. Gather necessary materials:** Shingles, underlayment, flashing, roofing cement, and safety gear. **3. Clear the area:** Remove debris from the roof to ensure a clean surface. **4. Isolate the leak:** Mark the location and contain interior damage. **5. Repair the leak (step-by-step):**

- **Shingle repairs:** Replace damaged shingles.
- **Flashing repairs:** Reattach or replace flashing.
- **Underlayment repairs:** Apply new material ensuring overlap.
- **Sealant application:** Use roofing cement on gaps.

**6. Inspect and test:** Perform a water test to confirm the fix.
**Additional Tips:** Prioritize safety. Consult a structural engineer for complex issues.

**GPT-5 Judgment**

The MPO response is clearly superior because it is more conversational, structured, and intuitively guided. It begins with an empathetic tone. The instructions are organized into clearly numbered, bolded steps with sub-bullets that logically progress from diagnosis to materials, repair, and verification.

**Gemini 3 Pro Judgement**

The MPO response is slightly better because of its superior organization and inclusion of a "Gather necessary materials" section. By grouping the repair steps into specific categories, it allows the user to easily navigate to the specific type of repair they need.

## Question:

*Can music help you fall asleep? What kind?*

**DPO Response**

Music can indeed play a significant role in promoting relaxation and helping you fall asleep faster. Research suggests that listening to calming music before bed can slow down your heart rate and reduce cortisol levels.

Here are some music genres and characteristics that can help you fall asleep: **1. Classical music:** Composers like Mozart, Chopin, and Bach. **2. Nature sounds:** Rainy sounds, ocean waves, or breezes. **3. Instrumental ambient:** Soft electronic instrumental tracks. **4. Lo-fi music:** Mellow, slow-paced hip-hop or jazz. **5. White noise:** Consistent sounds like static.

**Tips:** Keep volume low (20–40 decibels); choose consistent tempo; listen for 20–30 minutes before bed.

**MPO Response**

The eternal quest for a good night's sleep! Yes, music can be a powerful ally. Here are some top sleep-conducive music genres:

**1. Classical music:** Soothing pieces by Mozart, Chopin, etc. **2. Nature sounds with music:** Rain or ocean waves combined with soft melodies. **3. Ambient music:** Atmospheric soundscapes with minimal melodies. **4. Lo-fi music:** Mellow beats and soft piano. **5. Delta wave music:** Binaural beats to stimulate deep sleep. **6. Slow tempo music (60–80 BPM):** Lowers heart rate. **7. Acoustic or folk music:** Cozy, intimate atmosphere. **8. Chillout music:** Downtempo or Trip-Hop.

**Tips:** Maintain low volume; avoid screens; create a calm environment. Sweet dreams!

**GPT-5 Judgment**

The MPO response is better because it offers a richer, more engaging, and comprehensive experience. It opens with a warm tone and introduces scientifically-grounded options such as delta wave music and slow tempo (60–80 BPM) music, explaining their effects on heart rate and brainwaves.

**Gemini 3 Pro Judgement**

The MPO response is slightly better because it provides more specific and scientifically grounded recommendations. While both responses cover standard genres, the MPO response includes "Delta wave music" and specifically highlights the importance of BPM synchronization.

## Question:

*How do you get water in the desert?*

**DPO Response**

Accessing water in the desert can be challenging, but there are several effective ways to obtain it.
**1. Oases:** Natural springs or wells found in valleys. **2. Rainwater harvesting:** Collecting rainwater from surfaces or rooftops. **3. Groundwater wells:** Drilling or digging wells. **4. Surface water sources:** Rivers or streams (managed carefully). **5. Desalination:** Removing salt from seawater (energy-intensive). **6. Atmospheric water harvesting:** Using fog collectors. **7. Transportation:** Tanker trucks or pipelines. **8. Conservation and reuse:** Efficient irrigation and greywater reuse.

**MPO Response**

A life-sustaining question in one of the harshest environments on Earth! Getting water in the desert is crucial for survival.
**1. Rainwater harvesting:** Collecting rainwater using cisterns or tarps. **2. Oases:** Natural springs supporting unique ecosystems. **3. Wells and boreholes:** Accessing groundwater. **4. Desalination:** Converting seawater in coastal deserts. **5. Atmospheric water collection:** Using condensers or fog collectors. **6. Cactus and plant-based water storage:** Harvesting plants like aloe vera. **7. Carrying and transporting water:** Trucks or traditional animal transport. **8. Solar still:** A low-tech method using the sun to evaporate and condense water. **9. Government-provided infrastructure:** Pipelines and reservoirs. **10. Conservation and efficiency:** Making every drop count.

**GPT-5 Judgment**

The MPO response is better because it delivers a more comprehensive and engaging explanation. It opens with a humanizing hook and goes beyond conventional methods by including modern innovations (atmospheric harvesting) and survival techniques (solar stills, plant-based water), making it richer and more educational.

**Gemini 3 Pro Judgement**

The MPO response is better because it interprets the question more comprehensively, addressing both large-scale infrastructure and individual survival techniques. The inclusion of personal, low-tech methods like "solar stills" makes the MPO response much more useful for a potential survival situation.

