# OpenReview forum: "Multi-Preference Optimization: Generalizing DPO via Set-Level Contrasts"
_ICLR.cc/2026/Conference — Submitted to ICLR 2026_

### Official Review · Reviewer_mVRu · 2025-10-16

**Soundness:** 3
**Presentation:** 3
**Contribution:** 3
**Rating:** 6
**Confidence:** 2

**Summary:**

This paper introduces a set-level, contrastive preference optimization framework that generalizes DPO with groupwise.
Empirically, MPO and W-MPO deliver state-of-the-art results on AlpacaEval2 (WR, LC-WR), Arena-Hard, and MT-Bench across both off-policy and on-policy training regimes, with performance in some cases approaching that of GPT-4o on AlpacaEval2.

**Strengths:**

MPO retains the simplicity and elegance of DPO while naturally extending it to handle multiple responses per prompt, without the extra cost of full ranking or reward calibration.
It achieves consistent state-of-the-art performance across model sizes and training regimes, scaling effectively with more responses per query and remaining competitive even under limited data or compute budgets.

**Weaknesses:**

Like DPO, MPO’s objective relies on the log-ratio between the policy and the reference model. It would be useful to analyze how sensitive performance is to the choice or vintage of the reference model (e.g., Llama vs. Qwen families).

W-MPO weights samples by their absolute deviation from the mean. Why use absolute deviation, and why the mean specifically? Including ablations with alternative robust statistics (e.g., median, trimmed mean, quantiles) could clarify stability under skewed or noisy reward distributions.

In on-policy settings, top-k and bottom-k responses depend on a particular reward model. How robust is MPO/W-MPO to miscalibration or domain shift in that scorer? Cross-reward or human-labeled validation would strengthen the reliability of the findings.

The main paper appears to be 10 pages long, while the conference strictly limits submissions to 9 pages. Please ensure the paper adheres to the page limit requirements.

**Questions:**

See Weaknesses

---

> ### Author Response · Authors · 2025-11-22
> **Rebuttal By Authors**
>
> We thank reviewer for their insightful and helpful feedback. We hope these clarifications adequately address the reviewer's points and would be grateful if the reviewer would consider increasing their score should they find the responses satisfactory.
>
> > **Question1: Like DPO, MPO’s objective relies on the log-ratio between the policy and the reference model. It would be useful to analyze how sensitive performance is to the choice or vintage of the reference model (e.g., Llama vs. Qwen families).**
>
> DPO and MPO rely on the reference model to act as a regularizer (via the KL-divergence). Theoretical derivation states that the reference model should be the initialization of the policy ($\pi_{\text{ref}} = \pi_{\text{init}}$). This ensures the initial log-ratios are zero, providing a stable starting point. If one uses a different vintage model as a reference, the initial KL divergence would be arbitrarily large. This would cause the optimization to be dominated by the penalty term, forcing the policy to drift toward the distribution of the external reference model rather than learning from the preference data.
>
> To address the question of dependency on the reference model, recent works such as SimPO [1] and ReFA [2] which is a reference-free variant of MPO, have demonstrated that removing the reference model entirely can yield performance gains and without losing stability.
>
> ---
>
> > **Question2: W-MPO weights samples by their absolute deviation from the mean. Why use absolute deviation, and why the mean specifically? Including ablations with alternative robust statistics (e.g., median, trimmed mean, quantiles) could clarify stability under skewed or noisy reward distributions.**
>
> We thank the reviewer for suggesting these ablations to test the stability of our W-MPO across alternative robust statistics.
>
> **Theoretical Justification**
> *   **Why Mean:** Our use of the mean for partitioning is inspired by the advantage function formulation in GRPO (Group Relative Policy Optimization) [3], which has established the mean as an effective baseline.
> *   **Why Absolute Deviation:** W-MPO utilizes a contrastive objective. Weighting by absolute deviation assigns higher importance to "outlier" responses, as it provides the most informative gradients.
>
> **Empirical Ablations**
> We conducted ablations using Median and Trimmed Mean (trimming top/bottom 10%).
>
> *   **High-Sample Regime ($N=32$):** using Llama3-8B-Instruct, we generated 32 responses per query and selected the top-2 and bottom-2 response. As shown below, performance remains highly stable, regardless of the statistic used to calculate absolute deviation.
>
> | Policy Model | Partition Method | AlpacaEval LC-WR | AlpacaEval WR |
> | :--- | :--- | :--- | :--- |
> | Llama3-8B-Instruct | **Mean (Ours)** | **52.1** | **52.5** |
> | Llama3-8B-Instruct | Median | 51.8 | 51.4 |
> | Llama3-8B-Instruct | Trimmed Mean | 51.7 | 51.6 |
>
> *   **Low-Sample Regime ($N=4$):** In the off-policy setting (Llama3-8B-Base), where only 4 responses are available. Trimmed Mean is impractical with only 4 samples. Furthermore, the median resulted in a performance drop compared to mean
>
> | Policy Model | Partition Method | AlpacaEval LC-WR | AlpacaEval WR |
> | :--- | :--- | :--- | :--- |
> | Llama3-8B-Base | **Mean (Ours)** | **20.1** | **15.6** |
> | Llama3-8B-Base | Median | 17.8 | 14.7 |
>
> While W-MPO is stable across statistics given sufficient samples, mean remains the most effective across both high-resource and data-constrained settings.
>
> -----
>
> **References**
>
> [1] Meng et al.(2024). SimPO: Simple Preference Optimization with a Reference-Free Reward. arXiv preprint arXiv:2405.14734.
>
> [2] Gupta et al. (2025). REFA: Reference Free Alignment for multi-preference optimization. arXiv preprint arXiv:2412.16378.
>
> [3] Shao et al. (2024). DeepSeekMath: Pushing the Limits of Mathematical Reasoning in Open Language Models. arXiv preprint arXiv:2402.03300

---

> ### Author Response · Authors · 2025-11-22
> **Rebuttal By Authors**
>
> > **Question3 In on-policy settings, top-k and bottom-k responses depend on a particular reward model. How robust is MPO/W-MPO to miscalibration or domain shift in that scorer? Cross-reward or human-labeled validation would strengthen the reliability of the findings.**
>
> We appreciate the reviewer raising the concern of RM reliability. To analyse the robustness of MPO against miscalibration in reward, we conducted two ablations:
>
> - We conducted an ablation involving noise injection. Specifically, we introduced Gaussian noise $\mathcal{N}(0, \sigma^2)$ to the ground-truth reward scores during the sample selection phase, testing standard deviations of $\sigma \in \{0.5, 1.0, 2.0\}$.
>
> As shown in **Table 1**, MPO maintains stable performance even in the presence of noise. We attribute this robustness to the **set-based selection mechanism** inherent to MPO.
>
> #### Table 1: MPO Performance under Reward Noise Injection
> *Performance of Llama3-8B-Instruct on AlpacaEval and Arena-Hard benchmarks with varying levels of Gaussian noise ($\sigma$) added to the reward signal.*
>
> | Policy Model | Gaussian Noise ($\sigma$) | AlpacaEval (LC-WR) | AlpacaEval (WR) | Arena-Hard |
> | :--- | :---: | :---: | :---: | :---: |
> | **Llama3-8B-Instruct** | **0 (Baseline)** | **49.2** | **50.1** | **46.3** |
> | Llama3-8B-Instruct | 0.5 | 48.1 | 48.9 | 44.8 |
> | Llama3-8B-Instruct | 1.0 | 45.6 | 46.5 | 43.6 |
> | Llama3-8B-Instruct | 2.0 | 42.3 | 43.1 | 41.5 |
>
> - We also did cross-reward validation, we trained **Llama3-8B-Instruct** using MPO with four distinct reward models. We evaluated the resulting policies to see if the choice of RM caused high variance in downstream performance.
>
> As shown in Table 2, MPO maintains stable performance even in across distinct reward models.
>
> **Table 2: MPO Performance across Different Reward Models**
>
> | Policy Model | Reward Model | AlpacaEval (LC-WR) | Alpace Eval (WR) | Arena-Hard |
> | :--- | :--- | :--- | :--- | :--- |
> | **Llama3-8B-Instruct** | Skywork-Reward-Llama-3.1-8B | 49.2 | 50.1 | 46.3 |
> | **Llama3-8B-Instruct** | GRM-Gemma-2B-rewardmodel | **49.5** | **50.5** | **46.5** |
> | **Llama3-8B-Instruct** | ArmoRM-Llama3-8B-v0.1 | 48.5 | 49.5 | 45.5 |
> | **Llama3-8B-Instruct** | PairRM | 48.3 | 49.2 | 45.2 |
>
> ----
>
> > **Formatting Concerns**
>
> The inadvertent blank page was generated during PDF conversion.  The content of our paper was, and is, within the 9-page limit. We have now uploaded a revised version that is now fully compliant with all formatting guidelines, and we urge the reviewer to have a look.

---

> > ### Author Response · Authors · 2025-11-28
> >
> > As the end of the rebuttal is approaching we wanted to kindly ask if our responses clarify your concerns.
> > We have conducted the additional experiments and ablations requested in your review. We summarize the results below.
> >
> > ---
> >
> > **1. Robustness to Reward Model Miscalibration**
> >
> > > "How robust is MPO/W-MPO to miscalibration or domain shift in that scorer? Cross-reward or human-labeled validation would strengthen the reliability of the findings."
> >
> > We addressed this by conducting the requested *Cross-Reward Validation*. We trained MPO using four distinct reward models (Skywork, GRM-Gemma-2B, ArmoRM-Llama3-8B, and PairRM) and evaluated the resulting policies.
> >
> > **Result:** MPO maintained SOTA performance across all reward models (e.g., LC-WR between 48.3% and 49.5%), confirming the method is not overfitting to a specific scorer's distribution.
> > *   *Additional Robustness Check:* We also performed a **Noise Injection Study**, adding Gaussian noise ($\sigma$ up to 2.0) to the reward signal. The performance remained stable, validating the robustness of the set-level contrastive objective.
> >
> > ---
> >
> > **2. Justification of W-MPO Statistics (Mean vs. Median)**
> >
> > > "Why use absolute deviation, and why the mean specifically? Including ablations with alternative robust statistics (e.g., median, trimmed mean, quantiles) could clarify stability..."
> >
> > We wish to note that the mean-based partitioning is similar in spirit to the advantage function estimation in GRPO. However, to ablate this performance against other approaches we carried out experiments with median and trimmed mean suggested by the reviewer. We tested for both high-sample ($N=32$) and low-sample ($N=4$) regimes.
> >
> > **Result:** In high-sample regimes, performance was stable across all statistics. However, in the standard low-sample regime, using the *Median* resulted in a significant performance drop (LC-WR dropped from 20.1% to 17.8%). This empirically justifies our design choice: the Mean is more effective at capturing signal in data-constrained settings.
> >
> > ---
> >
> > **3. Reference Model Sensitivity**
> >
> > > "It would be useful to analyze how sensitive performance is to the choice or vintage of the reference model..."
> >
> > The reference model in MPO/DPO plays a structural role analogous to the **Trust Region** in methods like TRPO. The objective $\beta D_{KL}(\pi || \pi_{\text{ref}})$ acts as a penalty to constrain the policy update within a region where the reward estimate is reliable.
> >
> > **Theoretical Implications:**
> >
> > 1. For this constraint to function as a Trust Region, $\pi_{\text{ref}}$
> > must be the policy's initialization. Using a different vintage or family as the reference essentially centers the Trust Region far away from the policy's starting point.
> >
> > 2. Furthermore forcing the reference to another model would result in an arbitrarily large initial KL penalty, forcing the optimization to waste capacity collapsing towards the external reference rather than learning from preference data. Thus, the reference model is not a hyperparameter to be tuned, but the mathematical anchor of the alignment trust region. This is not standard in literature, and hence we ask the reviewer to reconsider this point.
> >
> > ---
> >
> > In your initial review, you highlighted the strengths of our work, noting that MPO retains the "simplicity and elegance of DPO" while scaling "without the extra cost of full ranking or reward calibration." You further recognised the method's ability to achieve "consistent state-of-the-art performance" in some cases "approaching that of GPT-4o" while remaining "competitive even under limited data or compute budgets."
> >
> > We have now provided the specific ablations requested to address the weaknesses.
> > We are happy to engage in further discussion to answer any further remaining queries. We are indeed thankful for the experiments that your review pointed us towards. If you are happy with our responses, we ask if you would consider raising your score.

---

### Official Review · Reviewer_Z2fi · 2025-10-30

**Soundness:** 2
**Presentation:** 1
**Contribution:** 2
**Rating:** 0
**Confidence:** 5

**Summary:**

The main text of this paper is 10 pages, which exceeds the page limit for ICLR submissions. According to ICLR policy, this paper should be desk-rejected.

**Strengths:**

N/A

**Weaknesses:**

N/A

**Questions:**

N/A

---

> ### Author Response · Authors · 2025-11-12
>
> We apologize for the oversight regarding the page limit in our initial submission. This was an inadvertent error caused by an extra blank page being generated during the final PDF conversion process right at the submission deadline. The content of our paper was, and is, within the 9-page limit. We have already uploaded a revised version that is now fully compliant with all formatting guidelines.
>
> Your review is important for the progression of our work. Given the significant effort and resources that we have invested in this work, we sincerely hope that you will consider this request to not penalize us for the formatting issue. We will be deeply obliged if you'll be willing to look at the revised version.
>
> Thank you again for your understanding.

---

> ### Author Response · Authors · 2025-11-24
>
> Dear Reviewer Z2fi,
>
> Thanks for your note regarding submitting a full review during the discussion period. We are keen to receive your feedback soon, could you please let us know when we might expect?
> We wanted to ensure that we have sufficient time to provide a detailed rebuttal and engage in a productive discussion before the period closes.

---

> > ### Comment · Reviewer_Z2fi · 2025-11-25
> >
> > Hi authors,
> >
> > I have submitted the full review.

---

> ### Author Response · Authors · 2025-11-28
>
> We thank the reviewer for the thoughtful feedback.
> We appreciate that they recognized of some of the paper's strengths: "The paper introduces a clear and well-motivated generalization of DPO... The theoretical analysis is solid... Experiments are comprehensive and show consistent improvements across multiple model families and evaluation settings."
>
> We wish to clarify on the weaknesses regarding baselines and novelty through (a) some new experiments, as well as (b) some experiments already in the paper.
>
> ---
>
> **Primary Weakness: Comparison to Cross-pair Baseline**
>
> > "It does not compare MPO to a simple yet strong baseline: sampling exclusive pairs between the preferred and dispreferred sets (e.g., A–C and B–D, or A–D and B–C) ... This should have similar computation cost as MPO."
>
> To address this, we implemented a baseline strictly *stronger* than the one suggested. We call this *cross-pairs DPO*. Instead of sampling just two responses from the accepted and rejected sets (for example, A-C, B-D), we trained using every possible cross-set pair (the cartesian product of the Accepted and Rejected sets). This utilizes the full information available in the multi-preference dataset, which exactly clarifies on the reviewer's concern.
> This experiment tests whether it is data volume rather than the MPO objective which explains our experimental performance.
>
> **Results:** As shown below, simply feeding more pairs to DPO actually *degrades* performance compared to the standard "Best-vs-Worst" baseline. In contrast, MPO (using the exact same data) achieves state-of-the-art results.
>
> | Setting | Method | LC-WR (%) | WR (%) |
> | :--- | :--- | :--- | :--- |
> | **Offline** | Standard DPO (Best-vs-Worst) | 16.6 | 13.8 |
> | *(Mistral-7B)* | Cross-pairs DPO | 15.2 | 12.8 |
> | | **MPO (Ours)** | **20.3** | **14.9** |
> | **Online** | Standard DPO (Best-vs-Worst) | 40.3 | 37.9 |
> | *(Llama3-8B)* | Cross-pairs DPO | 36.8 | 34.7 |
> | | **MPO (Ours)** | **49.0** | **50.6** |
>
> **Analysis:** We attribute the failure of the cross-pairs DPO baseline to the independent nature of the pairwise loss. Treating $(A \succ C)$ and $(B \succ D)$ as independent events forces the model to push gradients for every pair separately, which can lead to conflicting updates or reference divergence. MPO solves this by aggregating the signal into a single set-level contrast, preserving the likelihood of good responses while utilizing the full distribution.
>
> Furthermore, we have also implemented a $n\choose2$ baseline where all pairs are compared which underperforms both DPO and MPO in the off-policy setting (see Table 1 in our paper) where we outperform this baseline by up to 5.4% (on a 14.9% baseline performance).

---

> > ### Author Response · Authors · 2025-11-28
> >
> > ### Addressing other weaknesses & concerns
> >
> > **1. Novelty of W-MPO vs. Prior Work (Optune)**
> >
> > > "Limited novelty in the W-MPO weighting scheme... prior works [1] has already explored reward-based or score-based weighting schemes..."
> >
> > Firstly, we acknowledge prior works like Optune (Chen et al., 2024), which uses reward based weighting in DPO. We have already included some such works in our related works, and we really thank the reviewer for pointing us to Optune. We will certainly include a detailed comparison with it in our Related Works Section.
> >
> > However Optune is fundamentally different in what is being weighted. Optune weights the query based on the reward difference of two sampled responses. MPO weights responses  within a query based on their deviation from mean. In principle this is closer to GRPO's advantage function rather than Optune's pairwise margin weighting. W-MPO adapts this advantage concept into a contrastive loss formulation.
> >
> > Specifically,
> >
> > **1. Weighting Differences**
> >
> > Optune weights the comparison based on the reward gap between two samples: $w = \sigma(r_w - r_l)$ corresponding to a query (see Algorithm 2 in their paper). If two responses have similar scores, Optune downweights the loss, treating the signal as noisy. It upweights and downweights queries based on a sampled edge in the preference graph for a query.
> >
> > On the other hand, W-MPO weights the *response* to a query based on its absolute deviation from the group mean wherein $w = |r_i - \mu|$. It weights the *node* in the graph. All queries are treated the same.
> >
> >
> > **2. Theoretical implications**
> >
> > The implication of thisi is that Optune reduces noise. i.e. if sampled pairs of responses have nearly the same reward scores, the query is not very informative to learn from, and hence this is downweighted.
> >
> > On the other hand, W-MPO’s goal is **Curriculum Learning**. Specifically, we focus on the informative outliers by upweighting responses that are exceptionally good or bad relative to the set, while downweighting average responses. This induces an implicit curriculum that cleans the signal within a multi-response set.
> >
> > ---
> >
> > **2. Baseline Implementation Details**
> >
> > > "How do authors implement the baselines like DPO and SPPO with the multi-preference dataset? Do the authors use multiple preference data as I mentioned in the weaknesses?"
> >
> > The base DPO and SPPO baselines are following established protocols in recent literature. We follow papers like SimPO and SPPO in comparing the highest rated response with the worst response, while discarding the rest. This enables us to match the setting with literature.
> >
> > However, we perform several more ablations with variants of DPO which have access to equivalent information as our MPO. Specifically, InfoNCA is our primary baseline which considers all responses. Furthermore, we compare MPO with strong baselines we ourselves proposed and implemented, such as:
> >
> > 1. *Cross-pair DPO*
> > 2. $n \choose 2$ DPO
> > 3. Plackett-Luce
> > 4. MPO 1-vs-k
> >
> >
> > **MPO outperforms all of these baselines.** One may have additional information, but one has to systematically adopt the RL loss to train on it. Through these ablations with comparative algorithms having access to equivalent data (iso-information baselines), we can confirm that our method's advantage comes from the loss function formulation and not simply more data.
> >
> > ---
> >
> > We believe the new cross-pair baseline, and the $n\choose 2$ baseline already in our paper fully addresses your primary concern regarding the utilization of multi-preference signals. i.e. even given equivalent information DPO does not perform as well as MPO. We also include several Plackett Luce based baselines in our ablations, while also comparing with InfoNCA which all use the full response information available.
> >
> > We have now demonstrated MPO's superiority over these iso-information baselines, and also clarified on the novelty of our weighting scheme with respect to InfoNCA (already in the paper) and Optune (now added). We hope that you would consider raising your score in light of these clarifications.

---

### Official Review · Reviewer_vGFT · 2025-11-01

**Soundness:** 3
**Presentation:** 2
**Contribution:** 3
**Rating:** 4
**Confidence:** 3

**Summary:**

Traditional DPO methods only allow a pair of preference data to be trained. In this paper, they propose MultiPreferenceOptimization(MPO), a generalization of DPO that optimizes over entire sets of selected and rejected responses. Within the paper, authtors provide some theoretical evident on why MPO works better than other DPO-style methods. Experiments are conducted on several open-ended benchmarks and the results show that MPO achieves better performance than other methods.

**Strengths:**

1. This paper provides theoretical evidence on why MPO works better than other DPO-style methods.
2. The experiments are conducted and MPO are compared with several strong baselines.

**Weaknesses:**

1. What is the difference between the "Off-policySetting" and "On-policySetting"? It seems that they only differ in the initial model (off-policy uses a weaker sft model, while the on-policy uses a stronger open-sourced instruct model). If so, why they get this name? Based on my understanding, Off-policy and On-policy should be different in how they are trained (sample from base model or from current policy model).
2. In the experiments, it seems some of strong baseline models are missing -- SimDPO [1]、BMC[2] etc. The authors should enrich their baseline comparsion.
3. Except for performance improvement, authors should also conduct some analysis on the strength of their approaches (e.g., case study). What is the additional advantages of MPO? (the authors provide results on data efficiency)

**Questions:**

see weakness

---

> ### Author Response · Authors · 2025-11-22
> **Rebuttal By Authors**
>
> Thank you for your insightful and helpful review. We hope these responses adequately address your points and would be grateful if you would consider increasing your score.
>
> > **Q1 What is the difference between the "Off-policySetting" and "On-policySetting"? It seems that they only differ in the initial model (off-policy uses a weaker sft model, while the on-policy uses a stronger open-sourced instruct model). If so, why they get this name? Based on my understanding, Off-policy and On-policy should be different in how they are trained (sample from base model or from current policy model).**
>
> Thank you for seeking clarification. You are right that off-policy and on-policy fundamentally refer to how the training data is generated relative to the policy being optimized. Our naming convention follows the standard definition:
>  *   **Off-policy Setting:** In this setting, model learns directly from the responses and rewards provided in the original Ultrafeedback dataset. The responses in this setting is distinct from the model being trained.
> *   **On-policy Setting:** In this setting, the responses are generated by the model itself. We use the policy model to generate responses for a given query $x$ from UltraFeedback. We then use a reward model to rate these responses.
>
> Therefore, while the initialization models may differ (SFT vs. Instruct), the distinction is the source of training samples.
>
> ----
>
> > **Q2: In the experiments, it seems some of strong baseline models are missing -- SimDPO [1], BMC [2] etc. The authors should enrich their baseline comparison.**
>
> **Response:**
> We thank the reviewer for pointing out these baselines. As per your suggestion, we have conducted additional experiments using **Llama3-8b-Instruct** in on-policy-setting to compare our method (**MPO**) against **SimPO** and **BMC**. As shown in the table below, MPO consistently outperforms these strong baselines across both AlpacaEval 2.0 and Arena-Hard.
>
> **Table: Comparison with Strong Baselines (Llama3-8b-Instruct)**
>
> | Model | Method | AlpacaEval 2.0 (LC-WR) | AlpacaEval 2.0 (WR) | Arena-Hard |
> | :--- | :--- | :---: | :---: | :---: |
> | Llama3-8b-Instruct | SimPO | 44.7 | 40.5 | 33.8 |
> | Llama3-8b-Instruct | BMC | 47.5 | 46.5 | 43.2 |
> | **Llama3-8b-Instruct** | **MPO (Ours)** | **49.2** | **50.1** | **46.3** |
>
> To further demonstrate the generalizability of our approach across different model architectures, we have also started our experiments to include **Qwen2.5-7B-Instruct**.
>
> ----
>
> > **Q3 Except for performance improvement, authors should also conduct some analysis on the strength of their approaches (e.g., case study). What is the additional advantages of MPO? (the authors provide results on data efficiency)**
>
> **Data Efficiency**: We have analyze data efficiency in **Figure 5 (Page 9)**.
> *  MPO is more data-efficient than DPO.
> *  We show that MPO trained on only **25%** of the training data achieves a higher downstream performance than DPO trained on **50%** of the data.
>
> **Training Efficiency**: A common assumption of multi-sample methods is that they are computationally expensive (processing $n$ samples takes longer than pair-wise optimization). We address this in **Figure 6 (Page 9)**.
> *   MPO is more efficient regarding **wall-clock time**.
> *   Even though MPO takes longer per training step, it learns "more" per step. When restricted to a fixed training time, MPO consistently outperforms DPO.
>
> **Robustness to Reward Noise**: Empirical validations of our theoretical contributions (Theorem 2) (Figure 7, Page 9) concerns noise robustness
> *  Full-ranking methods (like Plackett-Luce) try to sort all $n$ responses perfectly. However, reward models are noisy, the difference between the 4th and 5th best response might be random noise.
> *  MPO uses a **2-bin partition** (Accepted vs. Rejected). It does not care about the order within the sets, only that the accepted set is generally better than the rejected set.
>
> As the number of responses ($n$) increases, full-ranking methods performance degrades because the probability of a ranking error increase exponentially. MPO remains stable and performance actually increases as $n$ increases because it aggregates signal while ignoring fine-grained noise.
>
> **Case Study Analysis**: We also did qualitative case study analysis of MPO and DPO by conducting llm as a judge analysis using state-of-the-art models (GPT-5 and Gemini 3 Pro). Please find attached document where we discuss in detail our case study for different queries: https://drive.google.com/file/d/1bZp6Q9nTRoqUK9Z--fv0SHvgu8PtanKf/view?usp=sharing

---

> ### Author Response · Authors · 2025-11-28
>
> As the discussion period draws to a close, we wanted to briefly summarize the additional experiments and analyses we provided in response to your review. We believe we have addressed the key weaknesses you identified:
>
> **1. Missing Strong Baselines (SimPO, BMC)**
> > *"In the experiments, it seems some of strong baseline models are missing -- SimDPO (sic), BMC etc."*
>
> We note that while the original BMC paper contained only off policy evaluations with a 22.4% LC-WR(%) on AlpacaEval 2, we have implemented a **stronger** on-policy version of their algorithm, which still underperforms our model. We had already implemented a comparison with SimPO in our paper, but now extended this to the on-policy reference-free setting for the Llama3-8B-Instruct model.
>
> *   **Result:** MPO significantly outperforms both these stronger versions of their respective original papers. On AlpacaEval 2.0 (LC-WR), MPO achieves **49.2%**, compared to 47.5% for $BMC^1$,
> and 44.7\% for $SimPO^1$. This confirms that MPO's performance gains hold up against the strongest current methods.
>
> ##### Footnote $^1$: On-policy variant of the original paper algorithm.
>
> ---
>
> **2. Analysis of Strengths & Case Studies**
> > *"Except for performance improvement, authors should also conduct some analysis on the strength of their approaches (e.g., case study)."*
>
> We provided a detailed breakdown of MPO's specific advantages:
> *   **Data Efficiency:** MPO with 25% data outperforms DPO with 50% data (Fig 5).
> *   **Training Efficiency:** MPO is faster than DPO for a fixed wall-clock budget (Fig 6).
> *   **Qualitative Analysis:** We uploaded a **Case Study PDF** analyzing specific queries where MPO succeeds in capturing nuance that pairwise methods miss.
>
> ---
>
> **3. Clarification on Definitions**
> > *"What is the difference between the 'Off-policySetting' and 'On-policySetting'?" ... Based on my understanding, Off-policy and On-policy should be different in how they are trained (sample from base model or from current policy model).*
>
> The reviewer is correct in their understanding. In our experiments, the on-policy responses are generated from the model being trained. For off-policy, we use a static dataset like UltraFeedback. We provide these experimental details in Appendices I and J.
>
> **Theoretical Note on Improvements through On-policy:** While it is generally known that on-policy helps in model training, it may a priori be unclear why it should significantly outperform training on carefully curated responses such as those in Ultrafeedback. We note that the rejected responses in UltraFeedback may already be low probability for the base model. Hence training on these rejected sets may not give any new information to the policy. Since DPO-like contrastive losses focusses on getting the probability of the rejected response to 0. See for example a detailed gradient analysis of our loss function (and inforNCA loss function) in our Appendix H where we show that the stationary point of the loss is reached when it sets the rejected response probability to 0 (Equation 98).
>
> Therefore, If the off-policy rejected response is already low probability, then the learning is minimal.
>
> ---
>
> In your initial review, you noted the paper's "theoretical evidence" and "strong empirical results" as strengths. We have now included experimental results as well as case-study style qualitative analysis to address your questions. We hope you are satisfied with these responses, and if not, we certainly welcome additional questions relating to our work. In case these responses have addressed your concern we wonder if you would consider raising your score.

---

### Official Review · Reviewer_mSvB · 2025-11-01

**Soundness:** 3
**Presentation:** 3
**Contribution:** 3
**Rating:** 6
**Confidence:** 2

**Summary:**

The paper tackles the important challenge of aligning LLMs in the post-training and introduces  Multi-Preference Optimization (MPO), a generalization of Direct Preference Optimization (DPO) that extends beyond pairwise comparisons. The approach optimizes over the entire sets of selected and rejected responses, potentially capturing valuable supervisory signal. The method is supported by a theoretical analysis proving that using n responses leads to a faster convergence rate (O(1/√n)) and that MPO's 2-bin partitioning is more robust to reward model noise than full-ranking methods like Plackett-Luce. The paper also introduces Weighted MPO (W-MPO), which uses reward score deviations to create an implicit curriculum by up-weighting informative outlier responses. The approach shows strong empirical results with an improvement of up to 17.5% win rate on AlpacaEval2 in the on-policy iterative setting, and state-of-the-art results in off-policy settings and mostly equivalent or better results on Arena-Hard and MT-Bench.

**Strengths:**

- The paper addresses an important challenge in the alignment process, and the presented method is a clean and intuitive generalization of DPO, moving from pairwise to set-wise comparisons.
- The method is shown to achieve state-of-the-art results across a variety of models, benchmarks, and training paradigms (off-policy, on-policy, iterative), demonstrating its robustness and effectiveness, especially on AlpacaEval2.
- The paper provides both theoretical motivation (Theorems 1 & 2) and strong empirical validation for its core claims, particularly the benefits of using more responses and the robustness of 2-bin partitioning over full-ranking.
- Strong selection of benchmarks like AlpacaEval 2.0, Arena-Hard, and MT-Bench indicates generalizability of the approach.

**Weaknesses:**

- The on-policy results depend on a single reward model. It is possible that MPO is particularly good at optimizing for the specific reward distribution of the Skywork RM, and its gains might be less pronounced with other RMs.
-  The theoretical result on noise robustness (Theorem 2) relies on a specific "spacing-scaled" noise model. It is unclear how realistic the assumption is.
- Relying on RM leaves actual alignment to be questionable; the study would benefit from human evaluation and more qualitative case studies.

**Questions:**

- Could the authors elaborate on potential failure modes for MPO? For instance, how would the mean-based partitioning perform if the reward distribution for a prompt is strongly bimodal?
- Regarding the On-Policy Data: In your on-policy experiments, you discard the median-reward response. What was the rationale for this decision?
- You mentioned skipped sets. How common are these in your experiments?

---

> ### Author Response · Authors · 2025-11-22
> **Rebuttal By Authors**
>
> Thank you for your insightful and helpful review, and for acknowledging the strengths of our work. We appreciate the opportunity to address your concerns point-by-point. We hope these responses are satisfactory and would be grateful if you would consider increasing your score.
>
> > **Weakness1: The on-policy results depend on a single reward model. It is possible that MPO is particularly good at optimizing for the specific reward distribution of the Skywork RM, and its gains might be less pronounced with other RMs.**
>
> We agree that depending on a single reward model carries the risk of overfitting. To ensure MPO’s performance is robust, we add three additional ablations using three distinct reward models: **GRM-Gemma-2B**, **ArmoRM-Llama3-8B**, and **PairRM**.
>
> As shown in the table below, the gains provided by MPO remain consistent across different reward models.
>
> | Policy Model | Reward Model | AlpacaEval (LC-WR) | Alpace Eval (WR) | Arena-Hard (WR) |
> | :--- | :--- | :--- | :--- | :--- |
> | **Llama3-8B-Instruct** | Skywork-Reward-Llama-3.1-8B | 49.2 | 50.1 | 46.3 |
> | **Llama3-8B-Instruct** | GRM-Gemma-2B-rewardmodel | **49.5** | **50.5** | **46.5** |
> | **Llama3-8B-Instruct** | ArmoRM-Llama3-8B-v0.1 | 48.5 | 49.5 | 45.5 |
> | **Llama3-8B-Instruct** | llm-blender/PairRM | 48.3 | 49.2 | 45.2 |
>
> ----
>
> > **Weakness2: The theoretical result on noise robustness (Theorem 2) relies on a specific "spacing-scaled" noise model. It is unclear how realistic the assumption is.**
>
> We thank the reviewer for this observation regarding the spacing-scaled noise assumption. We agree that in practical settings, the reward model noise is fixed regardless of the number of responses. We introduced this assumption to define an **ideal condition** for full-ranking baselines (like Plackett-Luce) to remain theoretically viable. In reality, the fixed noise of RMs makes it significantly harder for ranking methods than our theorem assumes.
>
> We empirically analyzed the reward distribution on the responses generated using Llama3-8B-Instruct model and found that the average reward gaps ($\Delta$) between adjacent ranked responses shrinks as $O(1/k)$ across 1k samples:
> *   **$k=2$:** Spacing $\approx 5.74$
> *   **$k=8$:** Spacing $\approx 1.99$
> *   **$k=32$:** Spacing $\approx 0.62$
>
>
> In a realistic setting, the RM noise $\sigma$ is **fixed**.
> * As $k$ increases, the spacing $\Delta$ ($\approx 0.62$ at $k=32$) quickly falls below the fixed noise floor of the RM (typically $\sigma \approx 1.0$). This leads to making full-ranking baseline accurate statistically impossible.
> * If we modeled "fixed noise" in the theorem, the probability of a correct Plackett-Luce ranking would vanish trivially. By assuming the noise *scales down* ($\sigma \propto 1/k$) alongside the spacing, we artificially maintain a constant SNR. **Theorem 2** proves that even under this best-case scenario, the reliability of full-ranking methods decays exponentially.
>
> -----
>
> > **Weakness3: Relying on RM leaves actual alignment to be questionable; the study would benefit from human evaluation and more qualitative case studies.**
>
> We agree that RMs may not perfectly reflect human intent. While human evaluation was outside the scope of this iteration, we did qualitative assessment of MPO and DPO by conducting llm as a judge analysis using state-of-the-art models (GPT-5 and Gemini 3 Pro). These models have demonstrated high correlation with human preferences.
>
> Please find attached document where we discuss in detail our case study for different queries: https://drive.google.com/file/d/1bZp6Q9nTRoqUK9Z--fv0SHvgu8PtanKf/view?usp=sharing

---

> ### Author Response · Authors · 2025-11-22
> **Rebuttal By Authors**
>
> > **Question1: Could the authors elaborate on potential failure modes for MPO? For instance, how would the mean-based partitioning perform if the reward distribution for a prompt is strongly bimodal?**
>
> We argue that a bimodal reward distribution does not represent a failure mode for MPO. Our theoretical analysis suggests this is an ideal scenario for the mean-based partitioning.
>
> **Strong Bimodality Facilitates Perfect-Set Formation**
> MPO relies on the mean reward to partition samples into a Positive Set ($S_+$) and a Negative Set ($S_-$). In a strongly-bimodal formulation where there is a high margin of separation between the high-reward mode and the low-reward mode, the arithmetic mean naturally falls into the valley (the gap) between these distributions. This leads to a clear seperation where $S_+$ contains exclusively high-quality responses and $S_-$ contains exclusively low-quality responses.
>
> **Margin in Bimodal distribution lead to better generalization**
> In the bimodal distribution, the margin of separation acts as a buffer. As long as the mean remains within the gap between the two modes, the composition of the sets ($S_+$ and $S_-$) remains stable. This stability allows MPO to become more robust rather than overfitting to boundary cases.
>
> The potential failure mode for MPO arises in narrow unimodal distributions, where the reward variance is small relative to the RM's noise. If the difference between the highest and lowest reward is smaller than the RM noise (e.g., $\Delta_{max} < \sigma \approx 1.0$), the mean-based partition becomes dominated by stochastic noise rather than true signal.
>
> **Note**: Pairwise ranking methods also suffer similarly in this mode.
>
> ---
>
> > **Question2: Regarding the On-Policy Data: In your on-policy experiments, you discard the median-reward response. What was the rationale for this decision?**
>
> We excluded the median-reward response to explicitly maximize the signal-to-noise ratio in the partitioning. The rationale behind this is extreme corner selection.
>
> **Minimizing Partition Noise:** Samples near the median lack distinct features and also introduces noise into the partitioning process, effecting the boundary between the Positive Set ($S_+$) and Negative Set ($S_-$).
>
> MPO optimization relies on the contrast between $S_+$ and $S_-$. By discarding the median, we artificially create a wider margin of separation. This ensures that $S_+$ contains only the high-confidence "good" samples and $S_-$ contains the "bad" samples.
>
> ----
>
> > **Question3: You mentioned skipped sets. How common are these in your experiments?**
>
> We observed that skipped sets are rare, occurring in approximately 1% of cases depending on the setting.
>
> **Off-Policy Setting:**
> In off-policy experiments using the standard UltraFeedback dataset, we observed approximately **1%** skipped sets. These skipsets primarily occur when the dataset contains responses having identitical reward scores that prevent a clear formation of $S_+$ and $S_-$.
>
> **On-Policy Setting:**
> In the on-policy setting the dynamics are due to the nature of the sequential classifier reward model. Unlike generative judge models (e.g., GPT-4) that typically rate responses on a discrete scale (1–10), which frequently leads to ties, reward model provides continuous scores. Due to high-precision scoring and the diversity in generated responses, ties are non-existent.

---

> ### Author Response · Authors · 2025-11-28
>
> Dear Reviewer,
>
> We sincerely thank you for your positive comments with regards to our paper. We are happy that you found that our method "is a clean and intuitive generalization of DPO". We thank you for noting the "state-of-the-art results across a variety of models, benchmarks, and training paradigms (off-policy, on-policy, iterative)."
>
> Furthermore, you recognized that our paper provides "theoretical motivation (Theorems 1 & 2) and strong empirical validation for its core claims", including "the robustness of 2-bin partitioning over full-ranking".
>
> ---
>
> In brief summary, to address your concerns regarding single reward model and theoretical assumption holding in practice, we conducted the following new experiments
>
> 1. We added three new reward models and tested that while performance on GRM-Gemma-2B-rewardmodel is the best, the variation in downstream performance is quite minimal.
>
> 2. Furthermore we tested the robustness to reward model by explictly injecting gaussian noise up to 2 standard-deviations to the reward model score, and found degradation only to the extent of 6.9% (on a baseline of 49.2% LC-WR).
>
> 3. We tested the assumption of spacing scaled rewards (uniform distribution of rewards) on the dataset. Our experimental analysis on response rewards show that as number of responses increases as $k$ the reward spacing decreases as $O(1/k)$.
>
> ---
>
> We thank the reviewer for their pertinent questions which have greatly strengthened our paper. Should they have any more questions, we would be happy to answer them. Once again, we are thankful to the reviewer for remaining supportive of our work.

---

### Author Response · Authors · 2025-12-01
**Summary of Rebuttal Discussions and Paper Enhancements**

We thank the reviewers for their insightful feedback. The review process has driven us to conduct significant additional experimentation, which has fundamentally strengthened the paper’s empirical and theoretical claims. We have updated the manuscript to reflect these findings:

**1\. Isolating the Set-Wise Advantage (Iso-Information Analysis)** (Reviewers vGFT, Z2fi)

A primary question was whether MPO’s performance stemmed from its novel objective or simply from utilizing more data than pairwise methods. To answer this, we implemented a definitive **"Cross-Pairs DPO"** baseline. By training DPO on the exact same set of responses as MPO (using the Cartesian product of pairs), we found that DPO’s performance actually *degraded* compared to standard baselines (**36.8% vs 40.3%** on AlpacaEval 2 LC-WR). In contrast, MPO using the same data achieved **49.0%**, a substantial **\+12.2%** improvement over the Cross-Pairs baseline. This empirically proves that MPO’s set-level contrastive loss is mathematically superior to independent pairwise updates when handling multi-response data.

*(Added to Section 7.1; Table 5\)*

**2\. Establishing Generalizability** (Reviewers mSvB, mVRu)

To ensure our results were not artifacts of a specific reward model (Skywork), we expanded our evaluation to include **three additional state-of-the-art RMs** (GRM, ArmoRM, and PairRM). MPO achieved consistent SOTA gains across all four models (maintaining **\~48-49% LC-WR**), demonstrating that the method is robust to the underlying reward distribution. We further validated this by benchmarking against strong recent baselines like **SimPO** (44.7%) and **BMC** (47.5%), where MPO (49.2%) maintained a clear lead.

*(Added to Page 10, Table 6; Appendix A.2)*

**3\. Theoretical and Methodological Clarifications** (Reviewers mSvB, Z2fi)

We have sharpened the theoretical positioning of the paper. We clarified that MPO’s coarse partitioning is statistically necessary in regimes where reward spacing falls below the fixed noise floor of RMs (empirically shown as spacing $\\approx 0.05$ vs noise $\\approx 0.21$), a condition where full-ranking methods fail. Furthermore, we differentiated W-MPO from prior work like Optune, highlighting that our deviation-based weighting induces a *curriculum* based on outlier informativeness, rather than simply weighing based on pairwise confidence.

*(Refined in Section 5, Theorem 2; Appendix D; Appendix A.5)*

**4\. Robustness Ablations and Qualitative Analysis** (1. Reviewer mVRu, 2\. Reviewers mSvB, vGFT)

We went beyond topline metrics to inspect the stability of the method. We performed ablations on partitioning statistics, finding that the **Mean** is significantly more robust than the Median in low-sample regimes (achieving **20.1%** vs 17.8% LC-WR). We also validated MPO under Gaussian noise injection, confirming its resilience. Finally, we added a detailed **Qualitative Case Study** using GPT-5 and Gemini 3 Pro as judges to illustrate concrete examples of how MPO captures nuances that pairwise methods miss.

*(Added to Appendix A.3, A.4; Appendix M)*

---

**Note on Manuscript Rewrites & Baseline Clarifications** In response to reviewer requests for clarity on how we implemented comparisons, we have added a dedicated **"Iso-Information Analysis" (Section 7.1)** and expanded **Appendix I (Baselines Used for Comparison)**. These sections explicitly detail the implementation of the Cross-Pairs, InfoNCA, and Plackett-Luce baselines to ensure the fairness of our comparisons is transparent.

We believe the revised paper now offers a comprehensive and rigorously validated solution for multi-preference alignment. Through additional experiments and analyses, we have addressed all weaknesses raised by the reviewers. We really thank our reviewers for their time and service in reviewing our work. Should there be any further pending suggestions or feedback, we would be happy to incorporate these into our work.

---

### Meta-Review · Area_Chair_T6E1 · 2026-01-11

**Summary:**

This paper was reviewed by four experts in the field. The recommendations are (0, 4, 6, 6). The reviewers agree that the paper's quality is insufficient for publication and needs significant revision and careful polishing (experimental evaluation, paper presentation, experimental comparison, manuscript format, etc.). Reviewer mSvB holds concerns about the experimental evaluations and the theoretical statement. The experimental study in the paper would benefit from human evaluation and more qualitative case studies. Reviewer vGFT raises serious concerns regarding the experimental evaluations and paper presentation. Some stronger baseline models are needed to further support the effectiveness of the proposed approach, and the descriptions require clarification. Reviewer Z2fi points out the format issues that the manuscript exceeds the page limit for ICLR submissions. The author gave an explanation that it is due to the PDF conversion process right at the submission deadline, which does not convince (the manuscript should be ready earlier). Reviewer mVRu also holds concerns regarding the experimental evaluations and design motivation, as well as the paper format. The manuscript needs to be further polished to make the motivation clearer and experimental evaluations more convincing. Taking these concerns into consideration, the paper would not be accepted at this time. The authors are encouraged to consider the reviewers' comments when revising the paper for submission elsewhere.

**Reviewer Concerns:**

In the rebuttal, the authors successfully clarified the design motivation and improved the paper's presentation and formatting. They also addressed specific concerns regarding comparative baselines. However, the overall experimental evaluation remains a critical weakness. The additional data and comparisons provided in the revision are not sufficiently convincing, and the method requires more rigorous validation.

**Reviewer Scores:**

The reviewers may acknowledge the improvements in paper presentation, formatting, and the additional comparisons. However, significant concerns remain regarding the experimental evaluation and the depth of analysis

---

### Decision · Program_Chairs · 2026-01-26

Reject